



# Spatial Patterns of Phosphorus Fractions in Soils of Temperate Forest Ecosystems with Silicate Parent Material

Florian Werner[1], Tilman René de la Haye[2], Sandra Spielvogel[2], Jörg Prietzel[1]

[1]Chair of Soil Science, Research Department Ecology and Ecosystem Management, Technical University of Munich, Emil-Ramann-Straße 2, 85354 Freising, Germany
[2]Soil Science Section, Institute of Geography, Faculty of Science, University of Bern, Hallerstrasse 12, 3012 Bern, Switzerland

*Correspondence to*: Florian Werner (florian.werner@wzw.tum.de)

**Keywords**: Cambisol, geostatistics, organic carbon, oxyhydroxides, pedogenesis, podzolization, P fractions

**Abstract.** The stage of pedogenesis is a crucial indicator describing phosphorus (P) distribution, but also governing spatial P distribution patterns. Here, we assessed spatial patterns of P fractions and major P binding partners (e.g. organic C, pedogenic Fe and Al minerals) in a geosequence to describe spatial and pedogenetic changes of P distribution and to identify mechanisms for these changes. We found, that the distribution of total P was generally best matched by the distribution pattern of organic P, both showing decreasing content from the top- to the subsoil. Inorganic P was mainly ascribed as bound in unweathered rock at all sites, but with decreasing importance in later stages of pedogenesis. The pedogenetically young soil at Bad Brückenau also showed adsorbed inorganic P in the topsoil, probably due to high mineralization of organic P. Soil organic matter (SOM)-sesquioxide-complexes, as well as Al and Fe oxyhydroxides were identified as main binding partners of organic P at all stages of pedogenesis. With depth, the correlations of various P fractions with SOM decreased, whereas those with pedogenic Fe and Al oxyhydroxides increased. The change of sorbent is due to the mobilization of first Al, and in later stages of pedogenesis, of Fe in the topsoil. Both metals and its oxyhydroxides ($Al(OH)_i$, $Fe(OH)_i$) probably formed strong complexes with SOM and therefore retained P in the pedon. Due to the heterogeneous P distribution, our results suggest a differing ecosystem P nutrition strategy at each of our sites: from acquiring inorganic P from weathered primary rock to minimizing loss of organic P by recycling. We argue that even in early stages of pedogenesis, P recycling is a major driver of ecosystem P nutrition, however not as important as in later stages. We conclude that the stage of pedogenesis in silicate soils, as e.g. visible in degree and state of podzolization, serves as predictor for plant and microbial P nutritional strategies.



## 1. Introduction

Temperate forest ecosystems, as all terrestrial ecosystems, rely on the availability of phosphorus (P) from the soil, which is related to site parameters like, e.g., soil pH, bedrock material, precipitation, and stage of pedogenesis. Even though many soils in central Europe are young, the P status of forests is often low or insufficient (Ilg et al., 2009; Jonard et al., 2015). Soil

acidification and N eutrophication, intensified by anthropogenic deposits, additionally reduces plant available P (Mohren et al., 1986). Strong P sorption, e.g. onto humic-mineral-complexes (Gerke and Hermann, 1992), by Al and Fe oxides and hydroxides (oxyhydroxides) (Sims and Pierzynski, 2005), or clay minerals (Violante and Pigna, 2002), increased formation of short-range order Al or Fe phosphate, as well as decreased mineralization of organic P may limit forest growth in many ecosystems (Richardson et al., 2004; Laliberté et al., 2012). In addition, Gerke (2010) showed that humic-mineral-phosphate

complexes may also increase plant available P and can account for 50 – 80 % of the P in solution. Studying P availability in soil therefore implies accounting for the distribution of different P proportions, in addition to the total P status in soil, as it is assessed by wet-chemical fractionation of soil, and/or by advanced spectroscopic/-metric techniques (Kruse et al., 2015).

Different P fractions exhibit different properties in soils (Sims and Pierzynski, 2005) and chemical P resources in soils change during pedogenesis (Walker and Syers, 1976). For example, soils accumulate organic P compounds during the first

500 years of pedogenesis, as described in chronosequence studies by Turner et al. (2007) in New Zealand and (Prietzel et al., 2013) in China/Switzerland. In another recent study, Turner et al. (2012) described a formation of an illuvial horizon in < 2000 years in a sand dune chronosequence in New Zealand. A current research program in Germany about ecosystem P nutrition uses a geosequence to study P resources in soils (Lang et al., 2016). The geosequence is characterized by a P status gradient, resulting from different bedrock and soil age. In addition to the P binding form, spatial heterogeneity of P has been

identified as an important factor controlling P availability for plants (Jackson and Caldwell, 1993), or for an alpine treeline (Liptzin et al., 2013). However, P depth distributions in most cases were assessed only unidimensional (e.g., Ferro Vázquez et al., 2014), not addressing the question of horizontal variation of P fractions in soils. Recent studies about carbon distributions in soils have started to include these questions (Spielvogel et al., 2014; Angst et al., 2016). However, the 2-D depth distribution of P fractions and important P binding elements in soils are still unclear. In addition, we do not know of a

study that has used a geosequence to describe the change of spatial distribution patterns of P fractions in soil as effect of pedogenesis. Here, we present high resolution distribution patterns of P fractions and major binding elements at the profile scale (i) to describe the spatial and pedogenetic changes of P distribution during pedogenesis, and (ii) to identify the most relevant factors and mechanisms for these changes.





## 2. Materials and Methods

### 2. 1. Soil sampling and sample preparation

Soil samples were taken from four German sites stocked with mature beech forests. One site is located near Bad Brückenau (BBR), Gauss-Krüger-coordinates: 3566195 E, 5579975 N, another near Freiburg (site: Conventwald [CON]), Gauss-Krüger-coordinates: 3422803 E, 5321010 N. The third site is located near Mitterfels (MIT), Gauss-Krüger-coordinates: 4564502 E, 5426906, and the last near Lüß (LUE), Gauss-Krüger-coordinates: 3585473 E, 5857057 N. All soils are formed from siliceous material and are classified as Cambisols (IUSS Working Group WRB, 2015). However, they differ in parent material (basalt, gneiss and Pleistocene sand), and in their contents of organic C, total P, and pedogenic Al and Fe minerals. Detailed information on site and soil properties is presented in Table 1, more detailed information for soil horizons can be found in Prietzel et al. (2016b).

((TABLE 1))

At each site, a soil profile was excavated at a representative location. Samples were taken from the mineral soil at every possible intersection of a 10 cm rectangular grid (Fig. 1) with a steel tube (ø 2 cm, depth 3 cm). Depending on profile depth and stone content, 56, 71, 61, and 68 (BBR, CON, MIT, and LUE) samples were taken. Within this grid, up to five smaller gridded nests (up to 6 samples with a distance of 3 cm, taken with the same steel tube) were included to improve geostatistical models (Webster and Oliver, 2008). Soil samples were dried at 60 °C for 48 h and subsequently sieved (< 2 mm). Sieved and finely ground subsamples were digested totally with HF/HClO$_4$ according to Prietzel et al. (2015) to analyze the contents of total P (P$_{tot}$), Ca (Ca$_{tot}$), Fe (Fe$_{tot}$), and Al (Al$_{tot}$) by ICP-OES (Vista-PRO Simultaneous ICP-OES, Varian Inc., Palo Alto, CA, USA). This digestion was favored over the *aqua regia* digestion to ensure digestion of silicate minerals and prevent underestimation of total P (Hornburg and Lüer, 1999; Prietzel et al., 2015). Total organic P (P$_{org}$) was determined using the ignition loss method of Saunders and Williams (1955), and total inorganic P was calculated by subtracting P$_{org}$ from P$_{tot}$. Total C and N contents were analyzed with an Elementar VarioEL CN analyzer (Elementar GmbH, Hanau, Germany). Because all soils were acidic and free of carbonate, total C content can be accounted for total organic carbon (OC). Contents of Al and Fe bond in pedogenic minerals were analyzed by extraction with bicarbonate-buffered dithionite-citrate solution (Al$_{di}$ and Fe$_{di}$) according to Mehra and Jackson (1958), as modified by Holmgren (1967). Contents of poorly crystalline Fe pedogenic minerals, as well as of Al(OH)$_3$, interlayer Al hydroxyl polymers, Imogolite, and Allophane, were assessed by extraction with acidic NH$_4$ oxalate solution (Al$_{ox}$ and Fe$_{ox}$) using the method of Schwertmann (1964). The respective amounts of oxalate-extractable P (P$_{ox}$) and dithionite-citrate-extractable P (P$_{di}$) were quantified in the same way, and also analyzed by ICP-OES. In addition, oxalate extracts were analyzed for orthophosphate P (P$_{ox.inorg}$) by colorimetry using the ascorbic acid method of Murphy and Riley (1962), as modified by John (1970). The difference between P$_{ox}$ and P$_{ox.inorg}$ was also assigned to organic P forms (P$_{ox.org}$). The C:P ratio was obtained by dividing OC contents by P$_{org}$ contents.



((FIGURE 1))

## 2.2. Statistical data analysis

The data were analyzed statistically with the software package *R*, version 3.2.3 (R Core Team, 2015). To characterize and compare the magnitudes of the spatial variation of the investigated soil properties among the four profiles, coefficient of

variations (CV) were calculated for all variables. Additionally, in order to specifically address the horizontal and vertical variation of the variables, a vertical and a horizontal CV were calculated ($CV_{ver}$ and $CV_{hor}$). These were the medians of the coefficients for every column and row, respectively, if the profile is imagined as a table consisting of rows (i.e. depth increments) and columns (left to right part of the profile). The proportion of horizontal CV by vertical CV ($CV_{hor/ver}$) served as a measure of variability in profile width and depth; i.e. values below 1 expressed a greater variation in vertical than in

horizontal direction of the profile, and vice versa. Correlation patterns among the studied variables were analyzed using Spearman's rank coefficient ρ. An upper profile region was defined by all sample points starting from the mineral soil surface to those of 20 cm depth. The middle profile region included points from 30 cm depth down to 50 cm depth, and the lower profile region contained points from 60 cm depth and reached until 70 cm (BBR, CON), 80 cm (LUE), or 90 cm (MIT;) of depth (Fig. 1). Additionally, factor analysis was performed on all samples of a given soil profile using *psych*

package, version 1.4.3 (Revelle, 2015). The *Optimal Coordinate* method was used to select the number of factors. For rotation *varimax*, and as factoring method *principal axis factoring* (PA) was determined. Factor scores were identified using *Thurstone* regression.

Data were analyzed for a depth trend before maps were computed. This was performed by testing the data values and the y-coordinates for correlation using the Kendall rank correlation coefficient and a scatter plot with a linear regression model. As

first step, the data were checked for normal distribution using the Shapiro-Wilk-Test (Shapiro and Wilk, 1965). When the assumption of normal distribution was rejected, the variogram estimator suggested by Cressie (1993) was used to build robust variograms. In all other cases the classical method of moments estimator (Webster and Oliver, 2007) was applied. All variograms were computed using the *geoR*-Package (ordinary kriging), version 1.7–5.1 of Ribeiro Jr. and Diggle (2001) and the *gstat*-Package (universal kriging), version 1.1-0 (Pebesma, 2004). Each variogram was calculated for the maximum

distance between all samples, half of the maximum distance and one third of the maximum distance. For each distance, variograms were computed for six through thirteen bins in each variogram. The three variograms with the lowest root mean squared error, as well as the three variograms with the best fit were stored. For map computation, ordinary kriging was used, as well as universal kriging which used the y-coordinates as an external trend to take a possible trend into account. Including the ordinary and universal kriging maps, an adequate, representative variogram, was selected visually to extract sill, nugget,

and range parameters for Bayesian Kriging which again was performed using *geoR* package. The *prior.control* parameters were set according to the variogram parameters. Afterwards, Bayesian Kriging was performed using a 1 cm prediction grid.



## 3. Results

### 3.1. Soil profile maps derived from Bayesian Kriging

((FIGURE 2))

The interpolated maps did not reveal a uniform distribution of P in any of the studied soils (Fig. 2). Total P showed high

contents in the upper part of the profiles and decreased systematically with soil depth at sites BBR and CON. At MIT, this fraction decreased less strongly with depth and at LUE $P_{tot}$ was distributed more patchily and characterized by a secondary maximum at 40 – 60 cm depth (LUE). Organic P generally showed the same trends, however, reached deeper at site BBR, and shallower at CON and MIT. At MIT, $P_{org}$ was additionally distributed more patchily than $P_{tot}$. The patchiness at LUE remained. At BBR, inorganic P was particularly enriched in the upper and lower parts of the profile, whereas it was enriched

in the upper and middle profile at CON and in the lower profile at MIT and LUE. For $P_{di}$, all soils except LUE showed highest contents in the upper profile (BBR, MIT, and CON), decreasing at about 20 cm (CON) to 40 cm (MIT) of depth. At LUE, no clear depth trend was found; however, $P_{di}$ showed a secondary maximum at 40 cm depth, but was distributed patchily. Especially at BBR, oxalate-extractable P showed a similar pattern depth as $P_{di}$. At the other sites, $P_{ox}$ was distributed more patchily, with a depth trend at CON and MIT. In addition, CON, MIT, and LUE showed secondary maxima

of $P_{ox}$ at about 40 cm depth. For MIT and CON, separation of the inorganic and organic oxalate-extractable P revealed a systematic change from a dominance of $P_{ox.org}$ in the uppermost 30 to 40 cm to a dominance of $P_{ox.inorg}$ below that depth. In all soils, the spatial distribution of $P_{ox.org}$ strongly resembled the distribution of $P_{org}$. Additionally, the spatial distribution of $P_{ox.inorg}$ resembled the distribution of $P_{inorg}$, however, at LUE, $P_{ox.inorg}$ was distributed more patchily.

Major P-bearing soil compounds also showed a heterogeneous distribution, i.e. SOM, represented by OC, and pedogenic Fe

and Al minerals with different crystallinity, represented by dithionite-extractable, as well as oxalate- extractable Fe and Al (Fig. 3). In particular, OC was enriched in the upper profile, particularly in the uppermost 10 cm, and showed a strong systematic decrease with depth. Total Fe generally increased with depth at BBR and MIT. In contrast, at CON and LUE, soil $Fe_{tot}$ contents showed a more patchily distribution and an enrichment zone at 40 – 50 cm depth. At BBR, CON, and MIT, contents of pedogenic Fe oxyhydroxides ($Fe_{di}$ and $Fe_{ox}$) systematically decreased with depth. LUE showed a pronounced

enrichment of $Fe_{di}$ and $Fe_{ox}$ at soil depths between 15 and 50 cm. The zone of maximum enrichment of $Fe_{di}$ (40 cm depth) was below the zone of maximum enrichment of $Fe_{ox}$ (20 cm). Moreover, compared to the other sites, $Fe_{ox}$ at LUE was distributed more patchily. Dithionite-extractable Fe was distributed more patchily than $Fe_{ox}$ at BBR, CON and MIT. Total Al was depleted in the upper profile at all sites, showing quite homogenous lateral distribution at BBR, MIT, and CON, but patchiness at LUE. A zone of maximum soil Al content at 40 – 60 cm depth was prominent at sites BBR, CON and LUE,

and at at 60 – 80 cm depth at MIT. Dithionite- and oxalate-extractable Al in general did not differ markedly in distribution. At BBR, CON, and MIT they showed a general decrease with depth. This general trend was superimposed by a upper profile depletion of $Al_{di}$ and $Al_{ox}$ which got more pronounced and deep-reaching in the sequence BBR < CON < MIT, associated



with formation of an enrichment zone at 20 cm depth (BBR, CON) or 40 cm (MIT). At LUE, a band of patched pedogenic Al minerals was found from a depth of 30 cm to 60 cm, with patches of maximum $Al_{di}$ and $Al_{ox}$ contents at 40 cm depth.

((FIGURE 3))

The C:P ratios were largest in the upper profile and decreased with depth at all sites (Fig. 4). At BBR, the ratio ranged from 13.6 to 76.0, whereas at CON – showing a similar spatial distribution of C:P as BBR – the C:P ratio ranged from 32.6 to 445.4. At MIT, no significant depth trend was detected and the C:P ratio was highest in the upper profile (213.7), but largely showing ratios below 100. At LUE, the profile was divided at about 40 cm depth. In shallower depth we found a depth trend of a C:P ratio ranging from about 150 (40 cm) to over 1300 (topsoil). Below 40 cm depth, the C:P ratio was predominantly below 150.

((FIGURE 4))

### 3.2. Coefficient of Variation analysis

In nearly all cases, CV analysis revealed a larger vertical than horizontal variation of all studied soil properties (Tab. 2). Major P-bearing soil compounds, as well as P fractions, showed largest total variation predominantly at LUE, and lowest total variation at MIT. The other two sites had intermediate CVs. Horizontal variation also was predominantly large at LUE, both for P fractions and all other compounds. The other sites did not show explicit trends. Vertical variation was often low at MIT, especially for organic P fractions, as well as for OC and pedogenic Al minerals. At LUE, we found the largest vertical variation for most P-bearing soil compounds, as well as for some P fractions, compared to the other soils. Due to the large horizontal variation at LUE, the proportion of horizontal to vertical CV ($CV_{hor/ver}$) was often large, compared to the other sites. Only OC showed a much larger vertical than horizontal variation at this site. In contrast, BBR showed small $CV_{hor/ver}$ for many P fractions due to a small horizontal variation.

((TABLE 2))

### 3.3. Correlation analysis

Spearman's rank correlation analysis of P fractions with OC and total, dithionite-, and oxalate-extractable Fe and Al minerals often revealed significant results (Tab. 3). Partial correlations for every fraction were calculated, but generally did not reach significance. In general, correlation coefficients were larger at BBR and CON, than at MIT, whereas LUE in most cases showed the smallest coefficients. Total P was strongly correlated with OC (BBR, MIT, and CON), followed by $Fe_{di}$, $Fe_{ox}$, and $Al_{di}$. For LUE, only moderate correlations of $P_{tot}$ with $Fe_{tot}$, $Fe_{ox}$, $Al_{di}$ and $Al_{ox}$, and no correlation with OC were found. This structure was nearly repeated for $P_{org}$; only at LUE, OC replaced $Fe_{tot}$. Correlations of all P fractions with $P_{inorg}$ were generally weaker than with $P_{org}$. In contrast to $P_{org}$, $P_{inorg}$ was correlated only loosely with total Fe and Al at MIT, CON, and LUE. Only at BBR, $P_{inorg}$ also weakly correlated with OC and pedogenic Fe minerals. Additionally, at site MIT, $Ca_{tot}$ was significantly correlated with $P_{inorg}$ and $P_{ox.inorg}$. Dithionite- and oxalate-extractable P showed similar results, as reported for $P_{org}$. However, coefficients of correlation were smaller for $P_{ox}$ than for $P_{di}$ at MIT and LUE. At the other two sites,



correlation of $P_{di}$ and $P_{ox}$ with OC, $Fe_{ox}$, $Fe_{di}$ and $Al_{di}$ was similarly large. Correlation with OC increased for $P_{ox}$, with respect to $P_{di}$ at MIT and CON. At the other two sites, correlations of $P_{ox}$ and $P_{di}$ with OC were both largest.

((TABLE 3))

In correlation analyses conducted for different soil depth increments, results often were not significant due to lower sample numbers. In the upper profile, OC was often highly correlated with $P_{tot}$ (BBR, CON, and LUE), organic P fractions (CON and LUE), as well as with $P_{di}$ and $P_{ox}$ (all sites). Oxalate-extractable Fe was less strongly correlated with the studied P fractions than OC, but generally more strongly than $Fe_{di}$. Correlations with pedogenic Al minerals ($Al_{ox}$, $Al_{di}$) appeared especially at LUE ($P_{tot}$, $P_{org}$, $P_{di}$, $P_{ox}$, and $P_{ox.org}$). As for the intermediate part of the soil profile (30 – 50 cm depth), OC was still an important correlation variable for $P_{tot}$ at BBR, MIT, and CON; additionally, also pedogenic Al and Fe minerals correlated significantly with $P_{tot}$. Only at LUE, $P_{tot}$ was correlated best with $Al_{tot}$ and $Fe_{tot}$, rather than OC. At this site, this pattern repeated for $P_{inorg}$. Generally, LUE was characterized by many insignificant correlations. The other three sites showed similar results for $P_{di}$, $P_{ox}$ and $P_{ox.org}$, namely high correlations with OC and pedogenic Al and Fe minerals. Only at BBR and CON, also correlations of $P_{org}$ with these fractions were found. In the lower profile the relevance of OC as correlation variable diminished for all P fractions. Total Fe, Al (BBR and CON), and Ca (MIT) took its place for $P_{tot}$ (partly also for $P_{inorg}$). Organic P was not correlated significantly with any of the studied variables at BBR and CON, but most strongly correlated with OC at MIT and with $Fe_{di}$ at LUE. Dithionite-extractable P was correlated strongly with $Fe_{di}$ at all sites and with $Al_{di}$ at BBR and MIT (moderately at LUE). Only at BBR, OC and $Al_{ox}$ were similarly correlated with this fraction. Oxalate-extractable P did not reveal uniform correlation patterns. Total Ca showed moderate correlations with $P_{ox}$ and $P_{ox.org}$ at LUE.

## 3.4. Factor analysis

Exploratory factor analysis revealed that two to three factors explained more than 65 % of the total variance of the data obtained for the four soils (Tab. 4). At BBR, the first principal axis (PA) had high loadings (> 0.7) in OC, $Al_{di}$, $Al_{ox}$, $Fe_{di}$, and $Fe_{ox}$. It thus can be termed *SOM-sesquioxide complex*. The second factor was loaded highly only with $P_{inorg}$ and $P_{ox.inorg}$ and can be termed *inorganic P*. As reported for BBR, also at MIT and CON, (i) PA 1 had high loadings in OC, $Al_{di}$, $Fe_{di}$, and $Fe_{ox}$ and can be termed *SOM-sesquioxide complex*, and (ii) all P forms, except $P_{inorg}$ and $P_{ox.inorg}$, were highly correlated with this factor. Different to BBR, at MIT and CON, PA 2 had high loadings on total contents of $Al_{tot}$, $Ca_{tot}$ and $Fe_{tot}$, but not $Al_{di}$ and $Fe_{di}$. It thus can be termed *unweathered bedrock*. At MIT and CON, the contents of inorganic P fractions were highly correlated with PA 2. In contrast to the other soils, at LUE sesquioxides and SOM loaded on independent factors. PA1, explaining most variance, had high loadings of $Al_{tot}$, $Al_{di}$, $Al_{ox}$, $Fe_{tot}$, $Fe_{di}$ and thus can be termed *sesquioxides*. Principal axis 2 had high loadings of OC, and this can be termed *SOM*. However, none of the studied P fractions was highly correlated with PA 1 or PA 2. Only Pox and Pox.org highly loaded on the third PA, which is why this factor was termed *available organic P*.

((TABLE 4))





## 4. Discussion

### 4. 1. Contents of SOM and pedogenic Al and Fe oxyhydroxides determine the spatial patterns of total P and different P fractions at the profile scale

Soil organic matter, represented as OC, was predominantly correlated with $P_{tot}$ and many other P fractions, especially in the topsoil. The interpolated maps of OC and $P_{tot}$ also show a striking similarity of the respective distributions of P and SOM. Organic P accumulates predominantly in the topsoil. For example, Syers and Walker (1969) reported that total P in the topsoil consisted of 63 % organic P after 10,000 years of pedogenesis from a wind-blown soil in a chronosequence study from New Zealand. In contrast to temperate, (proto-)spodic soils, pedogenetically younger Cambisols are characterized by a large pool of adsorbed $P_{org}$ species. It is known that non-crystalline minerals strongly influence turnover of SOM (Torn et al., 1997). Our results also show that pedogenic Al and Fe oxyhydroxides, represented as dithionite- and oxalate-extractable Al and Fe, were highly correlated with total P and predominantly organic P fractions, in particular in the middle and lower profile. At BBR, MIT, and CON, these compounds obviously were closely associated, and therefore combined as *SOM-sesquioxide-complex*, as revealed by factor analysis. This factor was highly correlated with various P fractions. In accordance, a positive relationship between the respective contents of Al, Fe and inositol phosphates was reported for soils by Vincent et al. (2012) in a study of boreal forest humus soils in Sweden. Pedogenic Al and Fe oxyhydroxides form by weathering of primary minerals in early stages of pedogenesis (Torn et al., 1997) and phosphate absorption to these minerals increases with decreasing pH (Goldberg and Sposito, 1984). It has often been reported that Al and Fe oxyhydroxides have a high capacity to adsorb $P_{org}$ (e.g., Celi et al., 1999; Yan et al., 2014; Prietzel et al., 2016a), as well as $P_{inorg}$ species (e.g., Parfitt, 1989; Violante and Pigna, 2002). At CON and MIT, our results indicate that adsorption of $P_{inorg}$ was negligible, probably due to competing adsorption of $P_{org}$. Thus, the amount of these oxyhydroxides, together with soil pH and the abundance of competing inorganic and organic ligands highly govern the availability of P (Hinsinger, 2001).

Originally, P is released by primary minerals, predominantly by dissolution of apatite (Walker and Syers, 1976), but can be preserved in small amounts as inclusion, e.g. in slowly weathering silicates (Syers et al., 1967). At sites MIT and CON, factor analysis revealed that inorganic P forms correlated highly with the factor *unweathered bedrock*, showing that the distribution of $P_{inorg}$ is connected with the distribution of slowly weathering rocks. In addition, at all sites there were correlations of P fractions with contents of total Al and Fe, especially in the subsoil. This indicates that, even at later stages of pedogenesis, unweathered primary rock bears some inorganic P forms, however, with decreasing impact on P nutrition with decreasing total P content (Walker and Syers, 1976). In a recent study of German forest soils performed with P $K$-edge XANES spectroscopy, Prietzel et al. (2016b) reported the presence of crystalline $AlPO_4$ in the bedrock of, e.g., BBR (24% $AlPO_4$) and MIT (35% $AlPO_4$). During pedogenesis, the adsorbed $P_{inorg}$ can be desorbed through displacement of other soil solution anions (Bhatti et al., 1998; Violante and Pigna, 2002). The spatial pattern of $P_{ox.inorg}$ at BBR, CON, MIT illustrates this desorption during early and intermediate stages of pedogenesis (Fig. 2): Easily available orthophosphate showed maxima in the topsoil (BBR), the upper (CON) and the lower subsoil (MIT) with advancing pedogenesis.



Organics on the other hand bind P often as relatively stable monoesters, up to 100 % as inositol phosphates (Condron et al., 2005). Active forms of organic P (e.g. orthophosphate diesters and labile orthophosphate monoesters) degrade faster than more stable forms, like inositol phosphates (phytates), which accumulate in soils (Condron et al., 2005). Therefore, mineralized P ("recycled P" or "biocycled P") becomes successively more important for plant and microbial nutrition during

pedogenesis (Turner et al., 2007). The inositol phosphates are retained in the soil by adsorption to clays and by SOM-sesquioxide-complexes or as precipitate with pedogenic minerals due to their high charge density, as reported by Condron et al. (2005). In addition, SOM in general, and thereby $P_{org}$, is (i) protected against mineralization by microorganisms when adsorbed to minerals, as recently reviewed by Han et al. (2016), or (ii) protected in the interior of soil aggregates (Lützow et al., 2006). Thus, P availability is controlled by desorption from, and by dissolution or mobilization of binding partners, i.e.

SOM, Fe and Al oxyhydroxides, and SOM-sesquioxide-complexes (Sims and Pierzynski, 2005).

Fractionation techniques have proven useful to answer questions about P bioavailability, e.g. as reported by Sherman et al. (2006) in a study about long-term acidification in spodic forest soils in Maine, USA. However, we also suggest analyzing the spatial patterns of P binding partners to explain the heterogeneity of P fractions in soil during pedogenesis. We conclude that the stage of pedogenesis and the associated spatial distribution of P binding partners is the crucial characteristic that

determines the spatial patterns of total P and other P fractions in our study.

## 4. 2. Effects of pedogenesis and podzolization on spatial patterns of total P and different P fractions at the profile scale

The soils from our sites showed different stages of pedogenesis. A rather early stage of pedogenesis was found at CON. In this soil, significant amounts of pedogenic Al minerals already have been mobilized from the topsoil and translocated

vertically, whereas Fe oxyhydroxides still showed the highest contents in the uppermost topsoil (Fig. 3), indicating incipient podzolization (Lichter, 1998). In addition, SOM contents decreased gradually from the topsoil to a depth of 20 cm (Fig. 3), and $CV_{hor/ver}$ values were often low (Tab. 2), depicting higher vertical than horizontal variation. It is well known that these patterns indicate an early stage of pedogenesis, i.e. an initial podzolization characterized by the dissolution of Al oxyhydroxides, but not Fe oxyhydroxides (e.g. Sauer et al., 2007; Turner et al., 2012). Additionally, the C:P ratios were

larger than 150 until a depth of about 40 – 50 cm (Fig. 4), indicating that the SOM was depleted in $P_{org}$ until this depth. Factor analysis revealed that all P fractions, except inorganic P fractions, highly correlated with the factor *SOM-sesquioxide-complex*, which accounted for 62 % of the total variation of the dataset (Tab. 4). The contents of $P_{tot}$ and $P_{org}$ were predominantly correlated with SOM, $Fe_{ox}$ and $Al_{di}$, esp. strongly in the topsoil (Tab. 3). This is backed by speciation results from P $K$-edge XANES spectra of Prietzel et al. (2016b) who assigned about 40 % of the total soil P to unbound organic P

and about 60 % absorbed to mineral phases in the CON topsoil. In the middle profile at CON, our results showed that Al and Fe oxyhydroxides had increased importance for P retention, probably due to absorption of P-rich SOM. This is in accordance with results of Prietzel et al. (2016b) who reported that more than 80 % of total P in the B horizon of CON was adsorbed to Al- and Fe-oxyhydroxides. In addition, also the mobilization of Al and Fe from primary rock was detected in our study, as



visible in the distribution of $Al_{tot}$ and $Fe_{tot}$, both showing content maxima at depths of 40 – 60 cm (Fig. 3). Enrichment zones of $P_{inorg}$, and $P_{ox.inorg}$ were also found at these depths (Fig. 2). This enrichment is probably due to crystalline $AlPO_4$ and $FePO_4$ which formed secondarily after weathering of primary apatites (Lichter, 1998), as indicated by our factor analysis results; both inorganic P fractions were highly correlated with the factor *unweathered bedrock*. This is again backed by Prietzel et al. (2016b) who reported that 17 – 25 % of total P is bound as crystalline $AlPO_4$ in the subsoil, and 65 % of the total P is bound as crystalline $AlPO_4$ in the bedrock.

Compared to CON, the Cambisol at MIT is characterized by more advanced podzolization. This is visible by bleached sand grains in the topsoil (Nechic subqualifier), and analytically detected by the distribution of pedogenic Al minerals, showing advanced dissolution of Al oxyhydroxides in the topsoil, compared to CON (Fig. 3). However, Fe oxyhydroxides were still stable in the topsoil. At intermediate stages of pedogenesis, the distribution patterns of many P fractions are characterized by a comparably homogeneous distribution. At MIT, also significant amounts of $P_{inorg}$, and especially $P_{ox.inorg}$, gradually increased from the top- to the subsoil. Calcium phosphate minerals (Ca-P) were found in the MIT bedrock by Prietzel et al. (2016b), bearing about two third of total bedrock P. The remaining third was identified as crystalline $AlPO_4$ (Prietzel et al., 2016b). They explained this large percentage by regolith formation which was associated with the transformation of apatite into $AlPO_4$ already at depths below 15m. However, the hill-top MIT soil lacks the rock outcrops that inhibited the podzolization of the hill-side CON soil, although both soils formed from regolith (Prietzel et al., 2016b). Our factor analysis and correlation analysis results both indicate that the inorganic P fractions in the MIT soil are predominantly bound in unweathered parent rock. The relevance of these P sources is backed by P speciation results of Prietzel et al. (2016b). They reported about a third of the total P to be bound to crystalline $AlPO_4$ in the subsoil at about 1 m depth, but did not detect Ca-P in the soil of the site, probably due to the low pH of the soil that leads to rapid dissolution of apatite (Lichter, 1998). The MIT soils showed this low pH also as a product of former large industrial, atmospheric S deposition, and ongoing large N deposition (Erkenberg et al., 1996). The pronounced SOM accumulation in the topsoil (Fig. 3) is probably a result of topsoil acidification, associated with a changed tree rooting pattern and a decrease in microbial activity (David et al., 1995). We assume that at MIT, mineralization of SOM also leads to a dislocation of $P_{org}$, as seen e.g. in decreasing vertical variation of OC and $P_{tot}$, in the presence of a secondary maximum of $P_{ox.org}$ at about 30 cm depth (Fig. 2), and in the homogenous distribution of the C:P ratio (Fig. 4). This is in accordance with results of SanClements et al. (2010). They showed that the dominant chemical fraction of P in a study of six watersheds in the eastern United States and Europe of soils under temperate forests was (i) associated with secondary Al and organic phases and (ii) responsive to experimental acidification. They suggested that bioavailability of P increases when soils are getting acidified due to increased mobilization of formerly protonated Al oxyhydroxides which are known to have a high anion adsorption capacity (Navratil et al., 2008; Kaňa et al., 2011). Probably, a significant fraction of P is released and translocated as colloidal P, as reported for two sandy model systems by Ilg et al. (2008). In our study, the dissolution of pedogenic Al minerals was visible in the interpolated profile maps (Fig. 3). As a result, many P fractions showed low total and vertical variability (Tab. 2). We therefore conclude that at MIT, acid-induced dissolution of Al oxyhydroxides reduces P variability throughout the profile, characterizing an



intermediate stage of pedogenesis. Only when acidification leads to dissolution of pedogenic Fe minerals – this is, when podzolization becomes visible – variability of P increases again.

Podzolization was most pronounced at LUE. Here, the sandy parent material did not provide abundant Al and Fe primary minerals that can retain P. In later stages of pedogenesis, P fractions are distributed highly variable, often showing distinct patches and spots with increased contents. At LUE, small amounts of pedogenic as well as primary Al and Fe minerals were enriched as a (proto-)spodic horizon below an eluvial horizon at depths below 10 cm. In this region, the contribution of poorly crystalline iron oxyhydroxide minerals ($Fe_{ox}$) was largest just below the eluvial horizon, whereas $Fe_{di}$, $Al_{ox}$ and $Al_{di}$ showed largest contents at greater depth (Fig. 3). In our study, $P_{tot}$ and $P_{org}$ accumulated in the illuvial horizon more pronounced then OC which indicates that the SOM in intermediate soil depths enriches in P during podzolization. This enrichment is also indicated by the change in C:P ratio at a depth of 40 – 50 cm (Fig. 4). It is known that (i) SOM and pedogenic Al and Fe oxyhydroxides form strong complexes (Sauer et al., 2007), and that (ii) Fe-SOM-complexes are less soluble and thus precipitate earlier than Al-SOM-complexes (Ferro Vázquez et al., 2014). We therefore attribute the P enrichment in the middle profile of LUE to the illuviation and retention of P-rich SOM which was dislocated from the topsoil, and retained in the illuvial horizons by absorption to first pedogenic Fe-, and in greater depth to pedogenic Al minerals. Kaiser (2001) showed that dissolved organic P (DOP) is enriched in the mobile hydrophilic fraction in subsoils of five acidic forest sites in Germany and highlighted that subsoils can effectively retain dissolved organic matter and thus DOP from these sites. This highlights the importance of Al and Fe oxyhydroxides for $P_{org}$ retention in the illuvial horizons of the LUE soil. Nonetheless, organic P also accumulated together with SOM in the topsoil and resulted in a significant rise of variability (Tab. 2). We detected the highest content of total P in the uppermost LUE topsoil (Fig. 2). This is in accordance with Vincent et al. (2010) who emphasized the importance of $P_{org}$ mineralization by microbial breakdown for plant nutrition from the litter and topsoil of P-deficient humus soils stocked with boreal forests. With increasing soil depth, Achat et al. (2009) suggested an increase of the importance of diffusive inorganic P in a temperate forest soil study. We also found highly diffused inorganic P (Fig .2), which resulted not only in high vertical, but also in high horizontal variability of inorganic P fractions (Tab. 2). Additionally, we found intermediate correlations of $P_{inorg}$ with $Al_{tot}$ and $Fe_{tot}$, but not with $Al_{di}$ or $Fe_{di}$, in the upper and middle profile (Tab. 3), indicating that inorganic P fractions can still be found in primary Al and Fe minerals. This implies that available inorganic P was not subjected to sorption. This is probably due to the generally low content of sorbents and the occupancy of sorbent exchange locations by $P_{org}$ fractions which also accumulated highly variably (Tab. 2). Therefore, we conclude that in later stages of pedogenesis, P fractions are distributed with large horizontal and vertical variability, leading to patches with P accumulation by SOM in the topsoil and by Al- and Fe-$P_{org}$-complexes in the illuvial horizons.





## 4.3. Case BBR: Spatial patterns of total P and different P fractions in a non-podzolized, acidic, oxide- and OC-rich soil

At BBR, pedogenesis is not advanced due to slow weathering of the basalt bedrock. Additionally, the hill-top location supports this persistence against weathering (Sauer et al., 2007). Phosphorus distribution patterns at BBR show a rare case of

an acidic soil in which podzolization is unlikely to occur in the near future due to the particular oxide-richness in the soil (Sauer et al., 2007). In the topsoil, soil pH had already decreased to 3.2 (Tab. 1), resulting in a depletion of calcium, its content increasing from the top- to the subsoil (not shown). Nevertheless, contents of Al- and Fe oxyhydroxides were still largest in the topsoil, compared to the middle and lower profile (Fig. 3). At this site, higher contents of $P_{inorg}$ were found in the upper and lower profile, whereas in the middle profile lower contents were detected (Fig. 2). Recent speciation results by

Prietzel et al. (2016b) showed that Ca-P minerals were found in the bedrock (67% of total P), in the fine earth in depths of 70 – 80 cm (49% of total P), and in the uppermost topsoil (22% of total P). Moreover, (Prietzel et al., 2016b) reported that 24 % of total P in the bedrock are bound as crystalline $AlPO_4$. The higher $P_{inorg}$ content in the subsoil is therefore likely due to unweathered apatite and $AlPO_4$ which is included in weathering-resistant minerals (Syers et al., 1967). The Ca-P in the topsoil, however, could neither be explained by our correlation results, nor our factor analysis, where solely inorganic P

fractions correlated to the second factor (Tab. 4). We assume that significant amounts of $P_{org}$ fractions were mineralized in the topsoil, which resulted in a considerable input of $P_{inorg}$ and subsequent $P_{inorg}$ sorption. Our results indicate that this biocycled $P_{inorg}$ was adsorbed by SOM, probably via Al or Ca bridge cations, and/or by Fe hydroxides – also indicated by the topsoil pattern of $P_{ox.inorg}$ (Fig. 2). Prietzel et al. (2016a) showed in sorption experiments that SOM, in buffered acidic conditions, retained 68 % of $P_{inorg}$, and 89% of $P_{org}$, when saturated with Al. In addition, Fe oxyhydroxides adsorbed 88 % of

$P_{inorg}$, and 89% of $P_{org}$ (Prietzel et al., 2016a). Recent speciation results by Prietzel et al. (2016b) showed that total P in the BBR Ah horizon consists of 41 % of amorphous $FePO_4$, 8 % P adsorbed by Fe oxyhydroxides and 40 % P adsorbed by Al-saturated SOM. Organic P fractions at BBR predominantly were accumulated in the topsoil and gradually decreased to the subsoil, with still relatively large contents at depths of 30 – 50 cm, compared to $P_{tot}$ (Fig. 2). The C:P ratios were generally small (Fig. 4), indicating high $P_{org}$ contents and significant SOM mineralization in the topsoil. In addition, European beeches,

which are the dominant tree species at BBR, are known to have abundant roots until depths of 40 cm which might contribute to the $P_{org}$ input at these depths (Schmid and Kazda, 2005; Spielvogel et al., 2014). Concerning its retention at depths of 30 – 50 cm, our correlation results indicate that $P_{org}$ fractions were predominantly adsorbed to Fe and Al oxyhydroxides (Tab. 3). A precipitation of P species seems unlikely due to the abundance of SOM that inhibits sesquioxide crystallization (Sauer et al., 2007). Our statement is backed by speciation results of Prietzel et al. (2016b) who found two third of the total soil P

absorbed to Al and Fe oxyhydroxides at BBR at depths between 10 and 70 cm. The early stage of pedogenesis at BBR is most evident in the variability of many P fractions, as indicated by our variation analysis (Tab. 2). The vertical CV was often much higher than the horizontal CV, suggesting a shallow weathered soil. Due to ongoing weathering of the basalt to Al- and Fe oxyhydroxides and 2:1 clay minerals (Eggleton et al., 1987) and due to the strongly acidic soil conditions, neither Al and



Fe oxyhydroxide, nor clay is likely to be translocated from the topsoil to the subsoil, which grants a strong resistance against podzolization.

## 5. Conclusion

In a recent publication, Lang et al. (2016) introduced a succession model of forest ecosystem nutrition, in which all our sites are included, based on the P status of the soils. They discriminated acquiring systems, in which plants and microbes obtain predominantly abundant, mineral bound P, and recycling systems, where mainly tight P biocycling sustains a poor, but sufficient P supply. In our study sites, the total content of P did not describe an adequate gradient of pedogenesis. It is known that soils from silicate bedrock in temperate forest ecosystems tend to podzolize in the course of pedogenesis. Among our sites, we therefore established a gradient of pedogenesis that accounts for their increasing stage of podzolization: BBR < CON < MIT < LUE. This scheme is able to adequately describe the zonation of P pools in acidic temperate soils on silicate parent material and it expands the model of Lang et al. (2016) by a geochemical description of the spatial distribution of P during pedogenesis (Fig. 5).

((Figure 5))

During initial stages of pedogenesis, apatite weathers and abundant P is taken up by plants and microbes, even in shallow depths. Also abundant, biocycled P is then retained by newly formed pedogenic oxyhydroxides in the topsoil, but also leaches in colloids and/or as DOP from the soil. In early stages of pedogenesis (BBR), soil acidification induces oxyhydroxide protonation and saturation of SOM exchanger sites with $Al^{3+}$ which leads to increased P retention in the topsoil. Reduced uptake of inorganic P is performed from larger depths, while seepage losses of P from the subsoil are still high. Progressive soil acidification results in (i) topsoil dissolution of first Al oxyhydroxides (CON), and later also Fe oxyhydroxides (MIT), and (ii) mobilization of oxyhydroxide-bound P. The mobilized P is re-adsorbed in illuvial horizons of the subsoil by newly formed SOM-sesquioxide-complexes, as well as Al and Fe oxyhydroxides. However, ongoing leaching of colloidal and dissolved organic P leads to increased P-depletion in the topsoil. Recycling of P from the illuvial horizon and, more importantly, from the topsoil, where organic P accumulates due to inhibited mineralization of SOM, then successively replaces P uptake as plant and microbial nutrition strategy. In later stages of pedogenesis (LUE), plants and microbes then predominantly rely on P recycling in the top soil (and O layer), followed by P recycling in the newly formed illuvial horizons, where highly stable SOM-sesquioxide-complexes still retain P. Thus, temperate forest ecosystems on silicate parent material tend to shift from relying on $P_{inorg}$ from weathered primary minerals to minimizing loss of P by tight $P_{org}$ recycling during pedogenesis.



**Acknowledgements**

We thank Sigrid Hiesch and Gertraud Harrington for their assistance in the laboratory work. This study was funded by the German Research Foundation (DFG), as part of the Priority Program SPP 1685, grant Pr 534/6-1.

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

**Tables**

25  **Table 1: General parameters of the four study sites Bad Brückenau (BBR), Conventwald (CON), Mitterfels (MIT), and Lüß (LUE).** mat: mean annual temperature, map: mean annual precipitation, pH in the topsoil, $mP_{tot}$: mean total P, $mCa_{tot}$: mean total Ca, mOC: mean organic Carbon, $mFe_{tot}$: mean total Fe, $mFe_{di}$: mean dithionite-citrate-extractable Fe, $mFe_{ox}$: mean oxalate-extractable Fe, $mAl_{tot}$: mean total Al, $mAl_{di}$: mean dithionite-citrate-extractable Al, and $mAl_{ox}$: mean oxalate-extractable Al.

| | BBR | CON | MIT | LUE |
|---|---|---|---|---|
| **soil profile** | | | | |
| **parent material soil type (WRB 2015)** | Basalt<br><br>Dystric Skeletic Cambisol | Paragneiss<br><br>Hyperdystric Skeletic Folic | Paragneiss<br><br>Hyperdystric Chromic Folic | Pleistocene glacifluvial sands<br>Hyperdystric Folic Cambisol (Arenic, |



| | (Hyperhumic, Loamic) | Cambisol (Hyperhumic, Loamic) | Cambisol (Humic, Loamic, Nechic) | Loamic, Nechic, Protospodic) |
|---|---|---|---|---|
| mat [°C] | 5.8 | 6.8 | 4.5 | 8.0 |
| map [mm] | 1031 | 1749 | 1299 | 779 |
| pH (CaCl$_2$) at 0 – 5 cm | 3.2 | 3.2 | 2.9 | 3.0 |
| mP$_{tot}$ [mg g$^{-1}$] | 2.08 | 0.37 | 0.72 | 0.06 |
| mOC [mg g$^{-1}$] | 42.66 | 33.09 | 29.39 | 9.37 |
| mCa$_{tot}$ [mg g$^{-1}$] | 11.74 | 1.34 | 7.93 | 0.97 |
| mFe$_{tot}$ [mg g$^{-1}$] | 82.61 | 42.13 | 47.69 | 4.54 |
| mFe$_{di}$ [mg g$^{-1}$] | 38.48 | 18.64 | 13.64 | 2.58 |
| mFe$_{ox}$ [mg g$^{-1}$] | 14.95 | 9.79 | 7.05 | 0.59 |
| mAl$_{tot}$ [mg g$^{-1}$] | 59.69 | 79.84 | 79.44 | 10.75 |
| mAl$_{di}$ [mg g$^{-1}$] | 7.81 | 6.65 | 5.84 | 0.56 |
| mAl$_{ox}$ [mg g$^{-1}$] | 7.66 | 7.30 | 6.59 | 0.53 |





| variable | BBR | | | | CON | | | | MIT | | | | LUE | | | |
|---|---|---|---|---|---|---|---|---|---|---|---|---|---|---|---|---|
| | $CV_{tot}$ | $CV_{hor}$ | $CV_{ver}$ | $CV_{hor/ver}$ | $CV_{tot}$ | $CV_{hor}$ | $CV_{ver}$ | $CV_{hor/ver}$ | $CV_{tot}$ | $CV_{hor}$ | $CV_{ver}$ | $CV_{hor/ver}$ | $CV_{tot}$ | $CV_{hor}$ | $CV_{ver}$ | $CV_{hor/ver}$ |
| OC | 73.7 | 18.6 | 76.2 | 0.24 | 91.4 | 25.0 | 94.9 | 0.26 | 49.7 | 17.4 | 48.4 | 0.36 | 201.9 | 33.4 | 181.5 | 0.18 |
| $Ca_{tot}$ | 31.9 | 18.8 | 31.4 | 0.60 | 11.9 | 7.7 | 11.2 | 0.69 | 11.6 | 6.8 | 12.1 | 0.56 | 19.8 | 12.3 | 17.4 | 0.71 |
| $Al_{tot}$ | 7.9 | 3.5 | 8.0 | 0.43 | 6.6 | 2.4 | 6.4 | 0.38 | 11.2 | 4.8 | 11.5 | 0.41 | 20.1 | 8.5 | 20.0 | 0.43 |
| $Al_{di}$ | 31.3 | 10.4 | 32.3 | 0.32 | 31.8 | 11.5 | 30.8 | 0.38 | 17.0 | 9.6 | 16.8 | 0.57 | 59.8 | 20.8 | 54.7 | 0.38 |
| $Al_{ox}$ | 21.7 | 9.7 | 21.9 | 0.44 | 26.2 | 10.1 | 26.1 | 0.39 | 18.0 | 8.3 | 19.6 | 0.42 | 65.1 | 20.3 | 49.6 | 0.41 |
| $Fe_{tot}$ | 6.9 | 3.5 | 5.3 | 0.66 | 5.5 | 3.9 | 5.7 | 0.68 | 4.9 | 4.1 | 4.4 | 0.94 | 31.7 | 14.7 | 30.9 | 0.48 |
| $Fe_{di}$ | 17.4 | 7.8 | 17.3 | 0.45 | 21.5 | 9.3 | 20.8 | 0.45 | 21.0 | 6.7 | 21.4 | 0.31 | 39.8 | 20.1 | 37.5 | 0.54 |
| $Fe_{ox}$ | 35.5 | 13.3 | 37.8 | 0.35 | 46.0 | 12.5 | 44.1 | 0.28 | 36.9 | 10.4 | 39.3 | 0.26 | 91.6 | 25.1 | 93.8 | 0.27 |
| $P_{tot}$ | 25.9 | 9.8 | 27.7 | 0.35 | 23.7 | 8.5 | 23.7 | 0.36 | 14.6 | 10.4 | 14.5 | 0.72 | 30.7 | 17.7 | 21.9 | 0.80 |
| $P_{org}$ | 33.9 | 12.2 | 35.4 | 0.34 | 44.9 | 16.5 | 43.6 | 0.38 | 32.7 | 19.6 | 26.4 | 0.74 | 59.6 | 30.4 | 56.3 | 0.54 |
| $P_{inorg}$ | 31.8 | 18.6 | 30.7 | 0.61 | 17.6 | 11.1 | 16.6 | 0.67 | 47.1 | 36.0 | 42.3 | 0.85 | 44.1 | 30.8 | 39.9 | 0.77 |
| $P_{di}$ | 52.0 | 12.0 | 54.0 | 0.22 | 28.4 | 10.1 | 29.1 | 0.35 | 24.5 | 7.9 | 25.5 | 0.31 | 64.3 | 58.0 | 64.7 | 0.90 |
| $P_{ox}$ | 34.9 | 7.7 | 37.2 | 0.21 | 42.9 | 18.5 | 42.5 | 0.43 | 14.5 | 11.0 | 14.2 | 0.77 | 91.3 | 50.7 | 87.5 | 0.58 |
| $P_{ox.org}$ | 52.7 | 14.8 | 57.1 | 0.26 | 101.6 | 41.5 | 102.4 | 0.41 | 32.0 | 12.9 | 31.7 | 0.41 | 91.8 | 65.9 | 90.8 | 0.73 |
| $P_{ox.inorg}$ | 31.5 | 12.6 | 19.7 | 0.64 | 29.2 | 16.5 | 30.4 | 0.54 | 38.2 | 13.9 | 39.6 | 0.35 | 147.7 | 155.3 | 147.5 | 1.05 |

**Table 2: Coefficient of Variation (CV) analysis results for the soils Bad Brückenau (BBR), Conventwald (CON), Mitterfels (MIT), and Lüß (LUE).** CV of all grid data points ($CV_{tot}$), the median of all horizontal and vertical CVs ($CV_{hor}$ and $CV_{ver}$), and the proportion of $CV_{hor}$ / $CV_{ver}$ are portrayed. Proportion values below 1 indicate a higher vertical than horizontal variation. Fractions are organic Carbon (OC), total, dithionite-citrate-extractable, oxalate-extractable Al, Fe and P ($Al_{tot}$, $Al_{di}$, $Al_{ox}$, $Fe_{tot}$, $Fe_{di}$, $Fe_{ox}$, $P_{tot}$, $P_{di}$, $P_{ox}$), organic and inorganic P ($P_{org}$, $P_{inorg}$), and organic and

5    inorganic oxalate-extractable P ($P_{ox.org}$, $P_{ox.norg}$).





| variable | full soil profile | | | | upper profile (0 – 20 cm) | | | | middle profile (30 – 50 cm) | | | | lower profile (>= 60 cm) | | | |
|---|---|---|---|---|---|---|---|---|---|---|---|---|---|---|---|---|
| | BBR | CON | MIT | LUE | BBR | CON | MIT | LUE | BBR | CON | MIT | LUE | BBR | CON | MIT | LUE |
| **P$_{tot}$** | OC 0.90 | OC 0.93 | OC 0.73 | Al$_{ox}$ 0.51 | OC 0.80 | OC 0.95 | ns | OC 0.78 | OC 0.86 | Fe$_{ox}$ 0.69 | OC 0.83 | Al$_{tot}$ 0.79 | Fe$_{tot}$ 0.76 | Fe$_{tot}$ 0.64 | Al$_{tot}$ 0.73 | Fe$_{di}$ 0.59 |
| | Fe$_{ox}$ 0.88 | Fe$_{ox}$ 0.89 | Fe$_{di}$ 0.61 | Fe$_{ox}$ 0.51 | Fe$_{ox}$ 0.54 | Fe$_{ox}$ 0.59 | ns | Al$_{ox}$ 0.74 | Al$_{di}$ 0.83 | OC 0.65 | Al$_{di}$ 0.65 | Fe$_{tot}$ 0.66 | Al$_{tot}$ 0.59 | Al$_{tot}$ 0.59 | Ca$_{tot}$ 0.68 | Fe$_{tot}$ 0.56 |
| | Fe$_{di}$ 0.83 | Al$_{di}$ 0.88 | Fe$_{ox}$ 0.58 | Fe$_{tot}$ 0.50 | ns | Fe$_{di}$ 0.44 | ns | Al$_{di}$ 0.62 | Fe$_{ox}$ 0.83 | Al$_{di}$ 0.64 | Fe$_{di}$ 0.64 | ns | ns | Al$_{di}$ 0.57 | OC 0.66 | ns |
| | Al$_{di}$ 0.80 | Fe$_{di}$ 0.80 | Al$_{di}$ 0.39 | Al$_{di}$ 0.46 | ns | ns | ns | Fe$_{ox}$ 0.55 | Al$_{di}$ 0.78 | Al$_{ox}$ 0.61 | Fe$_{ox}$ 0.55 | ns | ns | Al$_{ox}$ 0.56 | ns | ns |
| **P$_{org}$** | OC 0.92 | OC 0.94 | OC 0.77 | Al$_{ox}$ 0.50 | Al$_{tot}$ 0.49 | OC 0.88 | ns | OC 0.86 | Al$_{di}$ 0.81 | OC 0.77 | ns | ns | ns | ns | OC 0.73 | Fe$_{di}$ 0.52 |
| | Al$_{di}$ 0.91 | Fe$_{ox}$ 0.90 | Fe$_{di}$ 0.73 | Fe$_{ox}$ 0.50 | ns | Fe$_{ox}$ 0.66 | ns | Al$_{ox}$ 0.72 | Al$_{ox}$ 0.81 | Al$_{di}$ 0.75 | ns | ns | ns | ns | Fe$_{di}$ 0.49 | Fe$_{tot}$ 0.43 |
| | Fe$_{ox}$ 0.90 | Al$_{di}$ 0.86 | Fe$_{ox}$ 0.73 | Al$_{di}$ 0.48 | ns | Fe$_{di}$ 0.47 | ns | Al$_{di}$ 0.57 | Fe$_{di}$ 0.79 | Fe$_{ox}$ 0.68 | ns | ns | ns | ns | ns | ns |
| | Fe$_{di}$ 0.87 | Fe$_{di}$ 0.79 | Al$_{di}$ 0.43 | OC 0.42 | ns | ns | ns | Fe$_{ox}$ 0.43 | OC 0.77 | Al$_{ox}$ 0.66 | ns | ns | ns | ns | ns | ns |
| **P$_{inorg}$** | OC 0.40 | Al$_{ox}$ 0.51 | Al$_{tot}$ 0.66 | Fe$_{tot}$ 0.39 | OC 0.85 | ns | Fe$_{tot}$ 0.80 | Al$_{tot}$ 0.52 | OC 0.54 | ns | Ca$_{tot}$ 0.69 | Fe$_{tot}$ 0.60 | Al$_{tot}$ 0.70 | Fe$_{tot}$ 0.73 | ns | Al$_{ox}$ 0.41 |
| | Fe$_{ox}$ 0.37 | Al$_{tot}$ 0.45 | Ca$_{tot}$ 0.59 | Al$_{tot}$ 0.34 | Fe$_{ox}$ 0.53 | ns | ns | Fe$_{tot}$ 0.47 | Fe$_{ox}$ 0.49 | ns | OC 0.66 | Al$_{tot}$ 0.59 | Fe$_{tot}$ 0.69 | Al$_{tot}$ 0.67 | ns | ns |
| | Fe$_{di}$ 0.31 | Fe$_{tot}$ 0.38 | Fe$_{tot}$ 0.48 | Ns | ns | ns | ns | ns | Al$_{ox}$ 0.44 | ns | Al$_{tot}$ 0.62 | ns | ns | Fe$_{di}$ 0.63 | ns | ns |
| | ns | Al$_{di}$ 0.32 | ns | Ns | ns | ns | ns | ns | ns | ns | Fe$_{tot}$ 0.51 | ns | ns | Al$_{di}$ 0.58 | ns | ns |
| **P$_{di}$** | OC 0.98 | Fe$_{di}$ 0.88 | Fe$_{di}$ 0.96 | OC 0.50 | OC 0.82 | OC 0.95 | OC 0.81 | OC 0.70 | Al$_{di}$ 0.95 | Fe$_{di}$ 0.77 | Fe$_{di}$ 0.93 | Fe$_{di}$ 0.50 | Al$_{di}$ 0.87 | Fe$_{di}$ 0.66 | Fe$_{di}$ 0.87 | Fe$_{di}$ 0.67 |
| | Fe$_{ox}$ 0.96 | OC 0.84 | Fe$_{ox}$ 0.90 | Fe$_{ox}$ 0.42 | Fe$_{ox}$ 0.65 | Fe$_{ox}$ 0.73 | Fe$_{di}$ 0.76 | Al$_{ox}$ 0.56 | OC 0.93 | Fe$_{ox}$ 0.72 | Al$_{di}$ 0.86 | OC 0.47 | OC 0.80 | ns | Al$_{di}$ 0.79 | Al$_{di}$ 0.50 |
| | Al$_{di}$ 0.87 | Fe$_{ox}$ 0.84 | OC 0.90 | Fe$_{di}$ 0.31 | ns | Fe$_{di}$ 0.58 | Fe$_{ox}$ 0.74 | Al$_{di}$ 0.54 | Fe$_{ox}$ 0.92 | OC 0.53 | OC 0.84 | ns | Al$_{ox}$ 0.79 | ns | ns | Fe$_{tot}$ 0.48 |
| | Fe$_{di}$ 0.87 | Al$_{di}$ 0.82 | Al$_{di}$ 0.33 | Ns | ns | ns | ns | ns | Al$_{ox}$ 0.91 | Al$_{di}$ 0.52 | Fe$_{ox}$ 0.82 | ns | Fe$_{di}$ 0.78 | ns | ns | ns |
| **P$_{ox}$** | OC 0.94 | OC 0.92 | OC 0.73 | OC 0.39 | OC 0.78 | OC 0.93 | OC 0.59 | OC 0.85 | Al$_{ox}$ 0.89 | Al$_{ox}$ 0.71 | Fe$_{ox}$ 0.78 | ns | Fe$_{tot}$ 0.80 | Al$_{ox}$ 0.91 | Al$_{di}$ 0.62 | Al$_{di}$ 0.41 |
| | Fe$_{ox}$ 0.92 | Fe$_{ox}$ 0.90 | Fe$_{ox}$ 0.71 | Fe$_{ox}$ 0.36 | Fe$_{ox}$ 0.54 | Fe$_{ox}$ 0.67 | Fe$_{ox}$ 0.58 | Al$_{ox}$ 0.65 | OC 0.88 | OC 0.66 | OC 0.75 | ns | Al$_{tot}$ 0.64 | Al$_{di}$ 0.83 | Fe$_{di}$ 0.57 | Ca$_{tot}$ 0.41 |
| | Fe$_{di}$ 0.85 | Al$_{di}$ 0.88 | Fe$_{di}$ 0.70 | Ca$_{tot}$ 0.36 | ns | Fe$_{di}$ 0.51 | Fe$_{di}$ 0.44 | Al$_{di}$ 0.45 | Fe$_{ox}$ 0.85 | Fe$_{ox}$ 0.64 | Al$_{ox}$ 0.65 | ns | Fe$_{di}$ 0.56 | OC 0.57 | OC 0.42 | ns |
| | Al$_{di}$ 0.84 | Fe$_{di}$ 0.81 | Al$_{di}$ 0.30 | Al$_{ox}$ 0.32 | ns | Al$_{di}$ 0.42 | ns | ns | Al$_{di}$ 0.81 | Al$_{di}$ 0.59 | Fe$_{di}$ 0.61 | ns | ns | Fe$_{di}$ 0.54 | ns | ns |
| **P$_{ox.org}$** | Fe$_{ox}$ 0.96 | OC 0.97 | Fe$_{ox}$ 0.93 | OC 0.36 | Fe$_{ox}$ 0.54 | OC 0.94 | Fe$_{ox}$ 0.64 | OC 0.82 | Fe$_{ox}$ 0.93 | OC 0.85 | Fe$_{ox}$ 0.96 | ns | Al$_{ox}$ 0.94 | Al$_{ox}$ 0.60 | OC 0.71 | Ca$_{tot}$ 0.51 |
| | OC 0.94 | Fe$_{ox}$ 0.92 | OC 0.90 | Ca$_{tot}$ 0.33 | Fe$_{di}$ 0.44 | Fe$_{ox}$ 0.70 | OC 0.60 | Al$_{ox}$ 0.64 | OC 0.92 | Al$_{ox}$ 0.78 | OC 0.86 | ns | OC 0.88 | OC 0.54 | Fe$_{ox}$ 0.54 | ns |
| | Al$_{di}$ 0.90 | Al$_{di}$ 0.90 | Fe$_{di}$ 0.90 | Fe$_{ox}$ 0.31 | ns | Fe$_{di}$ 0.52 | Fe$_{di}$ 0.44 | Al$_{di}$ 0.46 | Al$_{ox}$ 0.92 | Al$_{di}$ 0.77 | Fe$_{di}$ 0.82 | ns | Fe$_{ox}$ 0.86 | ns | ns | ns |
| | Fe$_{di}$ 0.89 | Fe$_{di}$ 0.83 | Al$_{di}$ 0.29 | Al$_{ox}$ 0.26 | ns | ns | ns | ns | Al$_{di}$ 0.90 | Fe$_{ox}$ 0.65 | Al$_{di}$ 0.69 | ns | Al$_{di}$ 0.84 | ns | ns | ns |
| **P$_{ox.inorg}$** | ns | Al$_{tot}$ 0.73 | Al$_{tot}$ 0.79 | OC 0.32 | OC 0.71 | Al$_{ox}$ 0.57 | ns | OC 0.65 | ns | Ca$_{tot}$ 0.45 | Ca$_{tot}$ 0.46 | ns | Fe$_{tot}$ 0.79 | Al$_{ox}$ 0.85 | ns | ns |
| | ns | Fe$_{tot}$ 0.46 | Ca$_{tot}$ 0.56 | ns | ns | Al$_{di}$ 0.44 | ns | Al$_{ox}$ 0.43 | ns | Al$_{tot}$ 0.43 | ns | ns | Al$_{tot}$ 0.79 | Al$_{di}$ 0.82 | ns | ns |
| | ns | Ca$_{tot}$ 0.41 | Al$_{ox}$ 0.38 | ns | ns | ns | ns | ns | ns | ns | ns | ns | ns | Fe$_{di}$ 0.61 | ns | ns |
| | ns | ns | Fe$_{tot}$ 0.38 | ns | ns | ns | ns | ns | ns | ns | ns | ns | ns | ns | ns | ns |





**Table 3: Spearman's rank correlation coefficients of wet chemical P fractions from the full soil profile and different soil compartments at the study sites Bad Brückenau (BBR), Conventwald (CON), Mitterfels (MIT), and Lüß (LUE).** Fractions are total, dithionite-citrate-extractable, oxalate-extractable P ($P_{tot}$, $P_{di}$, $P_{ox}$), organic and inorganic P ($P_{org}$, $P_{inorg}$), and organic and inorganic oxalate-extractable P ($P_{ox.org}$, $P_{ox.inorg}$). These were correlated with organic Carbon (OC), total, dithionite-citrate-extractable, oxalate-extractable Al and Fe ($Al_{tot}$, $Al_{di}$, $Al_{ox}$, $Fe_{tot}$, $Fe_{di}$, $Fe_{ox}$) and ordered by correlation coefficient size (only the largest four values, if significant). Significance was assumed at p-values smaller 0.05; ns: not significant.





|  |  | BBR | | CON | | MIT | | LUE | | |
|---|---|---|---|---|---|---|---|---|---|---|
| factor contribution to total variance (%) | | PA 1 | PA 2 | PA 1 | PA 2 | PA 1 | PA 2 | PA 1 | PA 2 | PA 3 |
| | | 65 | 16 | 62 | 17 | 51 | 21 | 31 | 19 | 16 |
| | depth | -0.90 | -0.29 | -0.96 | 0.17 | -0.87 | 0.38 | 0.27 | -0.89 | -0.08 |
| | OC | **0.93** | 0.32 | **0.97** | -0.19 | **0.87** | -0.21 | 0.00 | **0.97** | 0.24 |
| | $Ca_{tot}$ | -0.72 | -0.43 | -0.65 | 0.27 | -0.10 | **0.79** | 0.46 | 0.53 | 0.18 |
| | $Al_{tot}$ | -0.57 | -0.32 | -0.30 | **0.86** | -0.52 | **0.79** | **0.87** | -0.22 | 0.09 |
| | $Al_{di}$ | **0.96** | -0.01 | **0.93** | -0.05 | **0.75** | 0.16 | **0.89** | 0.09 | 0.19 |
| v a r i a b l e s | $Al_{ox}$ | **0.87** | -0.20 | **0.74** | 0.30 | 0.25 | 0.39 | **0.81** | 0.07 | 0.29 |
| | $Fe_{tot}$ | -0.57 | -0.39 | -0.11 | 0.56 | -0.20 | 0.47 | **0.94** | 0.05 | 0.00 |
| | $Fe_{di}$ | **0.92** | 0.08 | **0.88** | -0.08 | **0.93** | -0.25 | **0.87** | 0.27 | 0.01 |
| | $Fe_{ox}$ | **0.94** | 0.24 | **0.95** | -0.12 | **0.91** | -0.30 | 0.60 | 0.61 | 0.21 |
| | $P_{tot}$ | **0.82** | 0.47 | **0.96** | 0.00 | **0.82** | 0.25 | 0.48 | 0.21 | 0.53 |
| | $P_{org}$ | **0.95** | 0.14 | **0.91** | -0.29 | 0.65 | -0.60 | 0.33 | 0.33 | 0.60 |
| | $P_{inorg}$ | 0.19 | **0.79** | 0.40 | **0.72** | -0.04 | **0.82** | 0.37 | -0.13 | -0.02 |
| | $P_{di}$ | **0.94** | 0.31 | **0.85** | -0.22 | **0.90** | -0.24 | 0.11 | 0.48 | 0.30 |
| | $P_{ox}$ | **0.88** | 0.41 | **0.96** | 0.01 | **0.86** | -0.02 | 0.08 | 0.16 | **0.95** |
| | $P_{ox.org}$ | **0.96** | 0.18 | **0.96** | -0.20 | **0.94** | -0.24 | 0.05 | 0.13 | **0.81** |
| | $P_{ox.inorg}$ | -0.01 | **0.85** | -0.23 | **0.82** | -0.71 | 0.42 | -0.10 | 0.30 | 0.31 |

**Table 4: Factor loadings of predictor variables at the study sites Bad Brückenau (BBR), Conventwald (CON), Mitterfels (MIT), and Lüß (LUE).** Predictors were depth, organic Carbon (OC), total, dithionite-citrate-extractable, and oxalate-extractable Al, Fe, and P ($Al_{tot}$, $Al_{di}$, $Al_{ox}$, $Fe_{tot}$, $Fe_{di}$, $Fe_{ox}$, $P_{tot}$, $P_{di}$, $P_{ox}$), organic and inorganic P ($P_{org}$, $P_{inorg}$), and organic and inorganic oxalate-extractable P ($P_{ox.org}$, $P_{ox.inorg}$). Number of principal axis (PA) selected by Optimal Coordinate method. Factor loadings above 0.7 are bold.



**Figures**

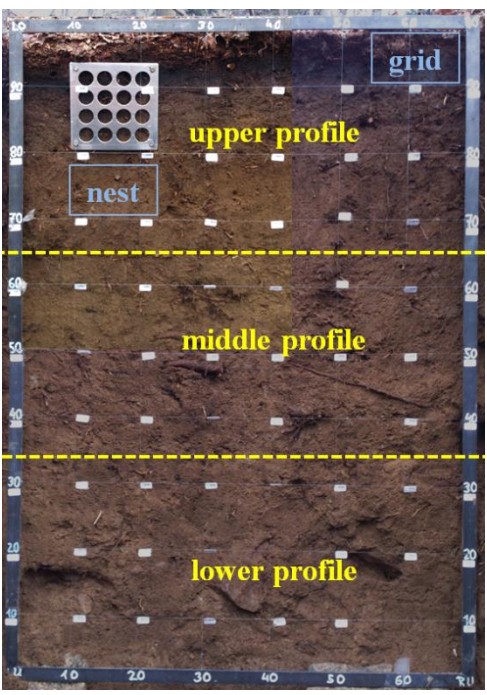

**Figure 1: Sampling grid with nest for smaller gridded sampling and scheme of soil region compartmentation.** The steel grid is 70 cm x 100 cm with intersections every 10 cm. For nested sampling, a steel plate with holes of a diameter of 2 cm and a distance of 3 cm per hole was used. The soil was divided in three regions: upper profile (0 – 20 cm depth), middle profile (30 – 50 cm depth), and lower profile (>= 60 cm depth).






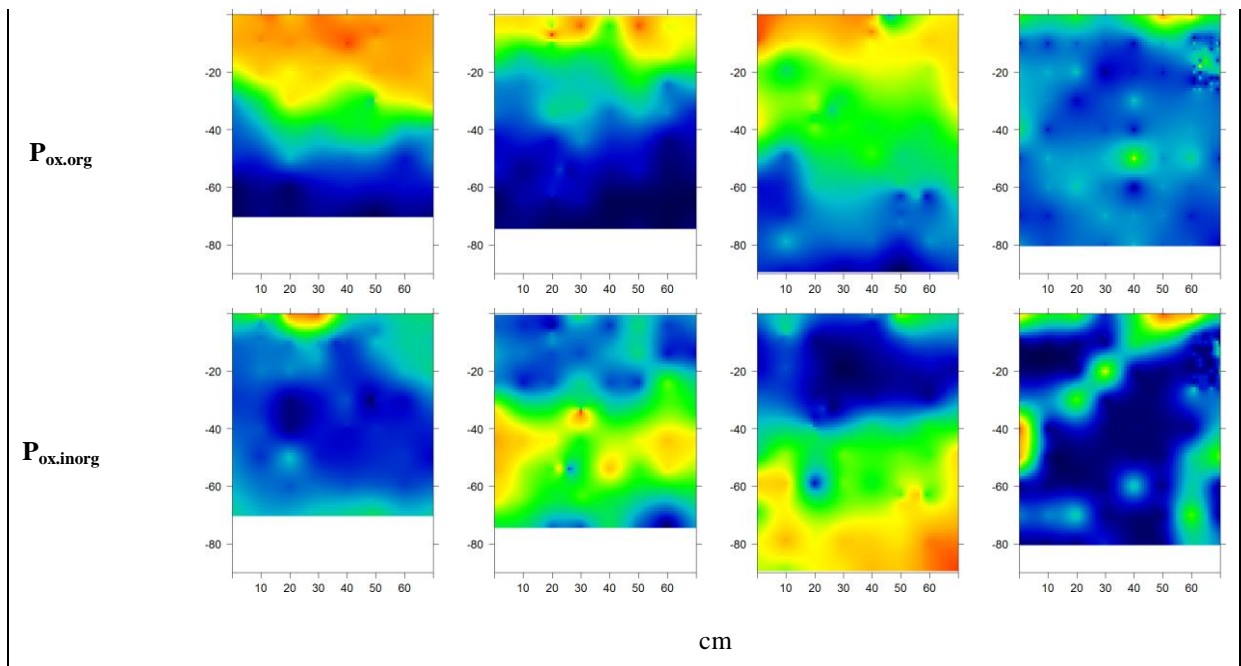

**Figure 2: Interpolated distribution patterns of total phosphorus (P) and different P fractions at the study sites Bad Brückenau (BBR), Conventwald (CON), Mitterfels (MIT), and Lüß (LUE).** Bayesian Kriging predictions (grid: 1 cm) of wet-chemical fractions of total P ($P_{tot}$), organic P ($P_{org}$), inorganic P ($P_{inorg}$), dithionite-citrate-extractable P ($P_{di}$), total oxalate-extractable P ($P_{ox}$), organic oxalate-extractable P ($P_{ox.org}$) and inorganic oxalate-extractable P ($P_{ox.inorg}$) are shown. Scales differ among the maps. Color ranges from low concentrations to high concentrations: black < blue < green < yellow < orange < red.







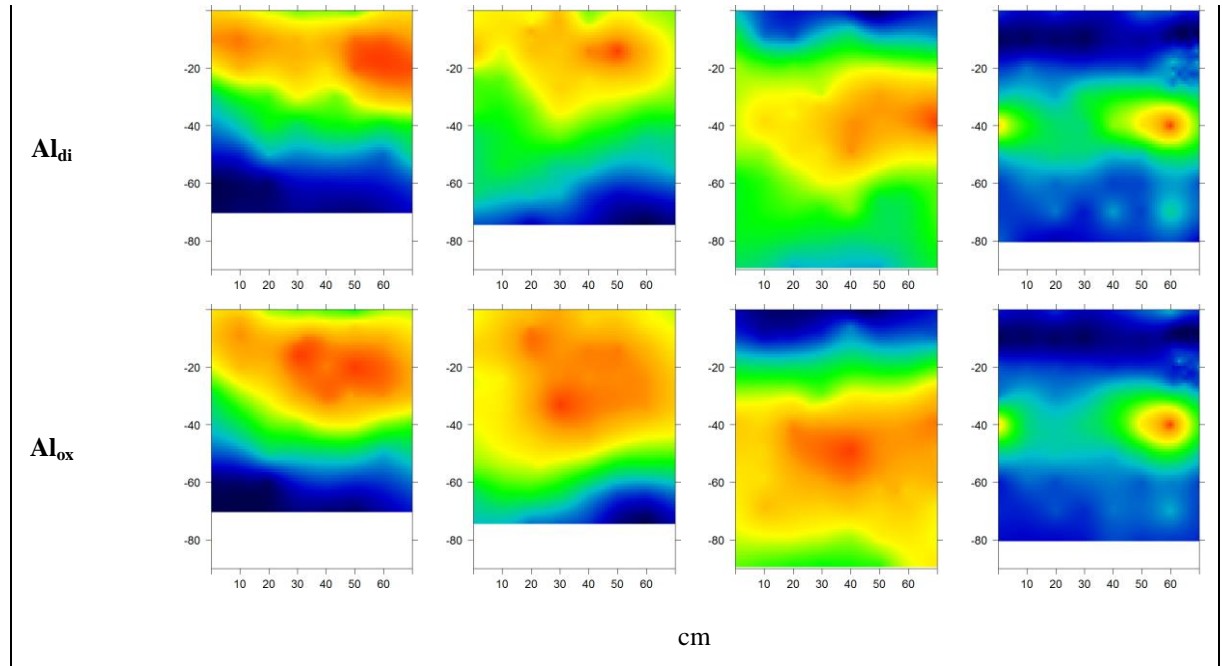

**Figure 3: Interpolated distribution patterns of organic C (OC) and Fe and Al bond in pedogenic minerals at the study sites Bad Brückenau (BBR), Conventwald (CON), Mitterfels (MIT), and Lüß (LUE).** Bayesian Kriging predictions (grid: 1 cm) of wet-chemical fractions of OC, total Fe ($Fe_{tot}$), dithionite-citrate-extractable Fe ($Fe_{di}$), oxalate-extractable Fe ($Fe_{ox}$), total Al ($Al_{tot}$), dithionite-citrate-extractable Al ($Al_{di}$), and oxalate-extractable Al ($Al_{ox}$) are shown. Scales differ among the maps. Color ranges from low concentrations to high concentrations: black < blue < green < yellow < orange < red.





**Figure 4: Ratios of organic C (OC) to organic P (P_org) at the study sites Bad Brückenau (BBR), Conventwald (CON), Mitterfels (MIT), and Lüß (LUE).** Bayesian Kriging predictions (grid: 1 cm) of the proportion OC/P_org. Note that the scale changes in every profile image.





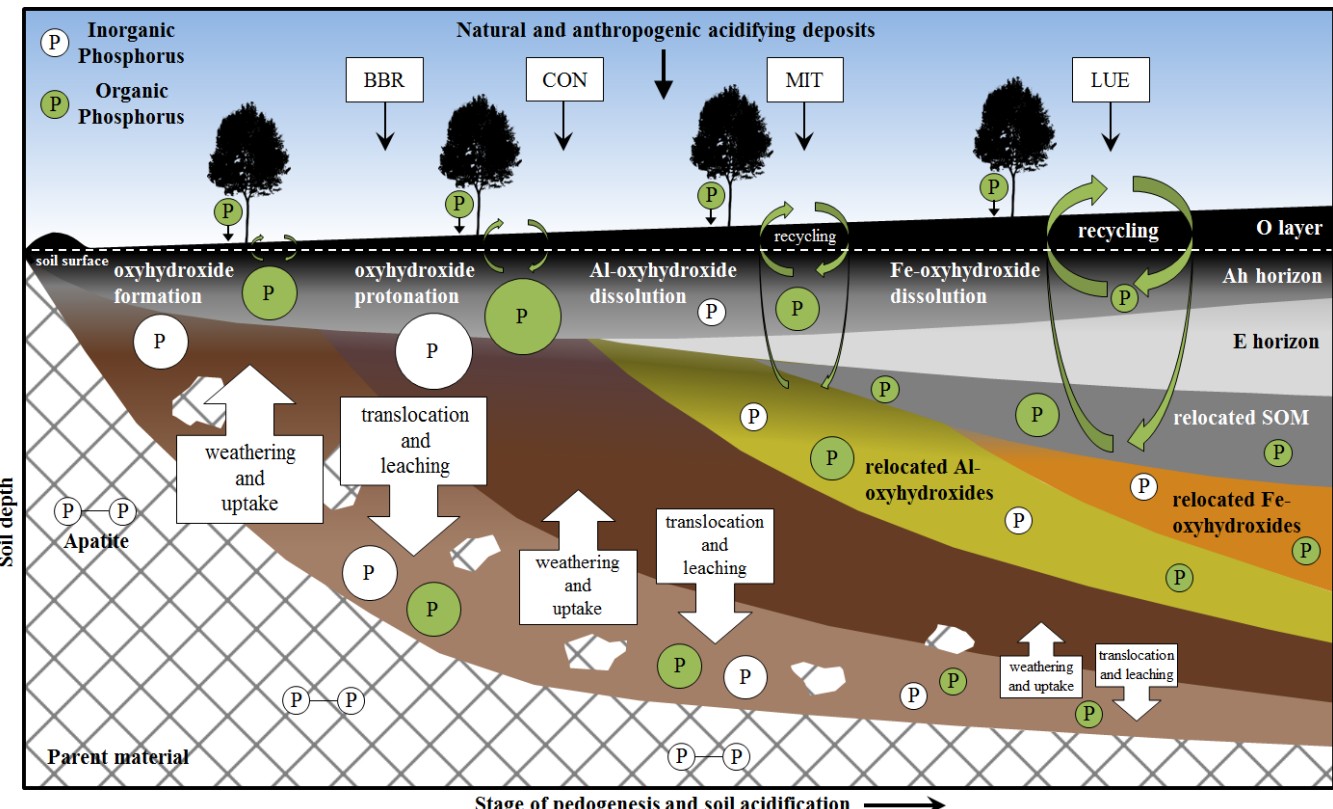

**Figure 5: Scheme of P distribution patterns during pedogenesis in temperate ecosystems, adapted from results of our study sites Bad Brückenau (BBR), Conventwald (CON), Mitterfels (MIT), and Lüß (LUE).**