# Peer review of "Spatial Patterns of Phosphorus Fractions in Soils of Temperate Forest Ecosystems with Silicate Parent Material"

_Biogeosciences, 2016_

## Referee Comment (RC1) · Anonymous Referee #1 · 10 May 2016

I am afraid I do not think this paper is suitable for publication, despite an enormous amount of work that went into it. I think this for three reasons, none of which are easily addressed, but which might be at least partially addressed with major revisions. I hope my comments below are useful in that effort.

1) The authors do not set up why the reader should care about small horizontal variation in soil properties (each of the many samples are taken from a 70x100cm grid). As far as I can tell there was only one grid per site. If the question is how do soils vary horizontally vs. vertically, this sampling design seems unlikely to be informative in any way that is ecologically relevant.

2) All of the krigging done basically reproduces a lot of what is know, there is strong

variation in soil properties (including C and P) with depth, and in very small areas less variation horizontally.

3) Any comparison of trends between the 4 sites, which differ in parent material, climate and elevation, is compromised by the issue of pseudo replication. At least from my reading there was only one soil pit excavated at each site, and soils were taken (in exhausting detail) from different parts of one pit. Thus comparison of the sites (e.g. effects of pedogenesis on basalt vs. gneiss) has only one true replicate for each site.

4) Even if one allows the samples from a single pit to count as true replicates, little can be inferred about the role of parent material, "pedogenic state" or anything else that varies between the sites because so much varies - there is almost 1000 mm/yr difference in rainfall between the sites, as well as very big differences in parent material, and differences in temperature.

Thus in my opinion the study is not appropriately set up to explore horizontal vs. vertical variation within a site, nor to study differences between sites. The major results (that there is large variation in soils with depth) is well known and the mechanisms for this have been explored for decades. The detailed work on P is interesting, but much of it has been published at a coarser scale by the same group.

I'm sorry I can't be more positive about this manuscript. I think a more robust introduction that sets up the questions and what is already known would go a long way to helping the reader, but I'm not sure that would over come the limitations of study design that I perceive here.

In general I think the introduction could be fleshed out. There should be some discussion of the different ways of assessing P forms (fractionation, NMR for organic P), and what their pros and cons are. If the ultimate goal is to understand pedogenic effects on P availability to organisms, there is a lot more literature that could be cited. If the goal is to see how P forms vary across this particular "geosequence" then I think more material is need to convince the reader that this a compelling question.

Along those lines, given that rainfall differs by almost 1000 mm/yr between sites, and there are different parent materials, it's hard to understand how these can be considered any sort of sequence. Rather it seems to me that it's four sites that have different soils, for a variety of reasons that can not be disentangled.

P1 L12 - what is a geosequence? Perhaps better to explain as you did in the introduction, as a series of sites that differ in P status due to differences in parent material and age.

L22 - I do not think that documenting different pools can be translated into an understanding of the pools from which P is acquired. There can be large pools of P that are not useful to organisms on short timescales.

L24 - Presumably the pedogenesis you refer to is all fairly early stage, and thus P availability is increasing as primary mineral P dissolves.

L25 - I am not sure what is novel about this result. The idea that soil development influences P forms and availability is quite old. What is the novel contribution of this work?

P2

L19 - How is soil age determined in this geosequence?

L22 - Why is the depth distribution important? Are rooting depths different across the geosequence? If so, how?

L24 - There is a great deal of literature on the distribution of P in soils, though less about P forms. Steven's work in the early 1970s in New Zealand had a wealth of information about P fractions with depth across the Franz Joseph and other chronosequences. The works from Hawaii (Crews et al, 1995 and subsequent) also has information. I believe Paul Selments has P fractions across the San Francisco Volcanic chronosequence, though I can't remember how much depth information he has. And of course Ben Turner has done a lot of work exploring organic N forms in myriad places and across

chronosequences.

L26 - I'm not sure from this intro how your data differ from Prietzel et al, 2016b. I'm sure they are different, but your introduction does not set up that difference very well. From my reading of that paper they also looked at P and what it's bound to across these sites.

P3

L8 - Basalt, gneiss and Pleistocene sand are very different parent materials, so I can understand why they would have wildly different P availability, and might host very different forms of P. However, there is no information given as to why the authors suggest these soils are of different age. In Prietzel et al., 2016b it is said that the 10 sites differ in lithology, but I can find no description of how they vary in soil age.

P3

L13 - This reads as if sampling at each site took place in a single 10cm square. But in truth it took place in a 70x100cm rectangle at every 10cm intersection. Is that correct? If so this text could be clarified?

On a more scientific note, why would a single grid be used to get a spatial variation, rather than many different locations? Some more explanation of why this approach was taken is warranted.

P4 - The motivation for the geospatial aspect of this paper is unclear to me. Until coming to the statistical analysis section of the methods, I had no idea there even was a geostatistical analysis, and even at this point in the paper I'm not sure what the goal of such a small spatial scale analysis is. I think this points to the fact that the introduction is so short that it does not really set up the motivation for the study or the questions as well as it needs to to bring the reader along.

P5

L4 - I don't think anyone would expect uniform distributions of P or any of its forms, and in general would expect higher concentrations of total P and organic P in the upper soils. So I'm not sure why this is a major result.

First paragaraph - I'm not sure why these results are being reported in this order. Is the major question about the difference between sites? This reads a bit like a long list without structure, and what is said here can be garnered from the table. It would be useful if the authors laid out guiding questions in the introduction, the methods used to answer those questions, and then structured the results in the same order.

L21 - The fact that organic carbon is concentrated in the upper soil has been reported for these exact soils, P8 -

L12 - This result has been observed many times before. I am left with wondering with what is gained by the extraordinary effort presented in this paper.

Figure 5 - It strikes me that this model would be very useful in the introduction. But I would also think that providing some justification for where you place the sites along the "pedogenesis" threshold is warranted. Since these are coming from different parent materials, where these are placed verge on the tautological. That is, one has a notion of how pedogenesis "should" progress, and then the soils are placed along the curve in a way that best fits the expectation. Given the wild differences in parent material, it seems like pedogenesis might take different tracks (see Vitousek and Chadwick 2013 for a similar idea as to how climate might influence pedogenic thresholds). Nevertheless, if this is your conceptual model, and you think your spatial analysis can inform the model, it might be useful to have this up front to guide the readers to your questions.

---

## Referee Comment (RC2) · Anonymous Referee #2 · 20 May 2016

This study presented high-resolution distribution patterns of P fractions and major binding elements at the profile scale (i) to describe the spatial and pedogenetic changes of P distribution during pedogenesis, and (ii) to identify the most relevant factors and mechanisms for these changes. They concluded that even in early stages of pedogenesis, P recycling is a major driver of ecosystem P nutrition, however not as important as in later stages and the stage of pedogenesis in silicate soils, as e.g. visible in degree and state of podzolization, serves as predictor for plant and microbial P nutritional strategies.

I found that their study is interesting, however, I suggest a major revision of the manuscript since the manuscript requires additional data/explanation to support their

discussion and conclusion and clarify some data/discussion. Below are my comments and suggestions.

1) They estimated "stage of pedogenesis" of their four soil samples. I wonder if it's reasonable to compare among soils with different parent material, moisture content or elevation. How were these four sites selected? Explanation of study sites and detailed soil property data are required. What does "mean" represent (i.e. entire soil depth)? What is the sample number (n=?)? P2 L9: I cannot find any data for LUE in the paper by Prietzel et al (2016b). In addition, the soil property data by Prietzel et al. (2016b) for BBR, CON and MIT were different from their data shown in Table 1. For instance, Prietzel et al. reported pH and TP of MIT (surface 0-2 cm) as 3.8 and 1.99 g P kg-1, respectively, whereas their data were 2.9 and 0.72 g P kg-1, respectively. Some soil properties such as texture and clay content should be added in Table 1.

2) They discussed P adsorption mechanisms in acidic soils, yet completely ignored clay content or/and types of clay present in each soil.

3) P5 L4:"The interpolated maps did not reveal a uniform distribution of P in any of the studied soils". This sentence is odd since no one expects uniform distribution of P in soils.

4) I found Figure 5 very confusing. The x-axis indicates, "stage of pedogenesis and soil acidification". However, according to Prietzel et al (2016b), the pHs of the soil samples are in the order of BBR (pH 3.1) > CON (3.6) > MIT (3.8) or according to their data, MIT (2.9) > LUE (3.0) > BBR = CON (3.2). Either way, they are not representing the stage of acidification. As I mentioned, I am not sure if they can compare the stage of pedogenesis among their soil samples.

5) Prietzel et al (2016b) estimated ∼65% of total P in the upper layer (0-10cm) of BBR was inorganic P, such as Ca(H2PO4)2 (11%), apatite (11%) and FePO4 (41%). Also ∼40% of total P in the upper layer of MIT was inorganic P, such as AlPO4 (18%) and FePO4 (22%). Yet, Figure 5 shows no inorganic P in the upper layer of BBR or MIT.

Any reason why?

6) P12 L8: How about adsorption of inorganic P onto clays in the upper layer? According to Prietzel et al. (2016b), the texture of BBR ((0-10cm) is silty clay.

7) P12 L24: Effects of root interaction on P transformation in soils should be included when thinking of distribution of forms of P. It will help to add approx. age of trees in each study site. I imagine that when they collected soil samples, they should have observed plant roots in different layers.

8) Figure 2 and 3: I liked the way they showed the distribution patters of TP and different P fractions. However, the range of proportion of each color is not clear. (i.e. what does the range high concentration of P represent?)

9) Table 3: I would like to see actual mean data in addition to the correlation.

10) In conclusion, I suggest adding some sentences to explain how their study can be useful to others and what might be the next step.

---

## Author Comment (AC1) · 16 Jun 2016

Answers to Reviewer 1:

"1) The authors do not set up why the reader should care about small horizontal variation in soil properties (each of the many samples are taken from a 70x100cm grid). As far as I can tell there was only one grid per site. If the question is how do soils vary horizontally vs. vertically, this sampling design seems unlikely to be informative in any way that is ecologically relevant."

In the introduction of the original version of the paper, we didn't explain the importance of small-scale variation of phosphorus (P) in soils well enough. The small-scale spatial

distribution of soil variables like pH, metal cations (esp. Ca, Fe and Al) and organic and inorganic ligands can be a crucial factor governing plant root activity (Hinsinger, 2001). In addition, local P depletion, mobilization, and transformation by plant roots results in small-scale ($\mu$m to cm) heterogeneity of soil P contents and P fractions (Hinsinger, 2001, Hinsinger et al., 2005). In our study, we performed a combined assessment of three issues: (1) P distribution by grid sampling (we addressed our research question by using a 70x100cm grid, supported by a nested sampling, s. manuscript P3 L13-16), (2) distribution of different P binding forms by wet-chemical analysis, and (3) distribution of explanatory variables. This assessment is of significant ecological relevance, because accumulation of P in soils and specific soil fractions has implications for plant P acquisition. We hypothesized that at sites with poor or intermediate P supply, spatial heterogeneity of P (fraction) contents results in (micro-)sites of P accumulation. At the moment no information can be found on the heterogeneity of physical and chemical soil properties of P content and P fractions at different scales. In the revised version of the manuscript we intend to clarify the aim and the novelty of our study by highlighting the (i) unknown issues of small-scale variation of P, (ii) our research hypothesis and (iii) how both are addressed in our study.

"2) All of the kriging done basically reproduces a lot of what is known, there is strong variation in soil properties (including C and P) with depth, and in very small areas less variation horizontally."

It is known that there is strong variation of C and P with depth and less variation horizontally. However, we do not know of any studies using P fractionation to address and explain small-scale P distribution patterns in soil and the relationship between the distributions of different P fractions with that of other soil variables (e.g. pedogenic oxyhydroxides). We therefore studied the distribution of total P and different P fractions to gain insight into spatial heterogeneity of P binding forms at different stages of pedogenesis (better: podzolization, s. next Author's answer). Geostatistics offered the possibility to produce high resolution maps, not only of total P and the P fractions, but

also of important variables for P binding. To the best of our knowledge such maps have not been published earlier.

"3) Any comparison of trends between the 4 sites, which differ in parent material, climate and elevation, is compromised by the issue of pseudo replication. At least from my reading there was only one soil pit excavated at each site, and soils were taken (in exhausting detail) from different parts of one pit. Thus comparison of the sites (e.g. effects of pedogenesis on basalt vs. gneiss) has only one true replicate for each site."

It is true that one soil profile per site was excavated and there is one replicate per site. We described the spatial and pedogenetic changes of P distribution at sites with differing pedogenetic age and identified the most relevant factors and mechanisms for these changes. In our manuscript, we used the term "stage of pedogenesis" to distinguish the "pedogenetic age" of our sites (referring to our sites as a "geosequence"). In the revised version of the paper we intend to replace these ambiguous terms using "stage of podzolization" instead to describe the pedogenetic age of our soils. The stage of podzolization is, of course, affected in combined effects of parent material, climate, elevation, and many more factors. The revised manuscript will clearly demonstrate that our study principally focused on P distribution in four soils (Cambisols) with different stage of podzolization of forest soils with siliceous parent material under temperate climate. We collected a large sample size. However, we did not statistically test for differences between parameter (e.g. OC and Ptot content) mean values of the four different soils. We discussed each site result independently and qualitatively set the site in a framework of "stage of pedogenesis". We did not claim to have replicated the sampling of our site, but we also did not statistically test for significant differences. We therefore argue that pseudoreplication is not a critical issue in our study because the differences between the sites are described and discussed qualitatively.

"4) Even if one allows the samples from a single pit to count as true replicates, little can be inferred about the role of parent material, "pedogenic state" or anything else that varies between the sites because so much varies - there is almost 1000 mm/yr

difference in rainfall between the sites, as well as very big differences in parent material, and differences in temperature."

We admit that it is audacious to develop a detailed description of the changes in P binding forms and P distribution with advancing stage of podzolization (manuscript Fig. 5) based on only one replicate per site. However, Fig. 5 and the comparison of the sites should be seen as a general conceptual model of relationships between stage of podzolization as identified by the distribution of pedogenic Al and Fe minerals and the distribution of different P forms and important P fluxes as derived from our results. We will do more to adequately reason the choice of our sampling sites. Despite the differences, all studied soils were formed from siliceous parent material under temperate climate and are stacked with European beech stands of similar age. The soils showed different pedogenetic age, as seen in stage of podzolization. Pre-studies especially found differing P contents in the beach leaves from these soils. The "sequence of stage of pedogenesis"-Figure is therefore rather a conclusion of our study of P (fraction) distribution, than a rationale of our work. We intend to clarify this issue in a thoroughly revised version of the manuscript.

"Thus in my opinion the study is not appropriately set up to explore horizontal vs. vertical variation within a site, nor to study differences between sites. The major results (that there is large variation in soils with depth) is well known and the mechanisms for this have been explored for decades. The detailed work on P is interesting, but much of it has been published at a coarser scale by the same group."

In the answer to comment (1), we argued that looking at the small-scale variation of P is expedient, ecologically relevant and provides novel information. Main results of our study not only comprise the distribution patterns of total and organic P and organic C (large variation with depth). The detailed work on the distribution of several P fractions and the relationship with the distribution of Alox/di and Feox/di in soils with differing P status ("stage of pedogenesis/podzolization") at smaller scales is essential to explain P distribution patterns in soil. We argue that the mechanisms governing small-scale P

distribution are affecting P acquisition of plants and soil microorganisms. The mechanisms are known; however, the detailed description of the spatial heterogeneity of P fractions in temperate forest soils is novel.

"I'm sorry I can't be more positive about this manuscript. I think a more robust introduction that sets up the questions and what is already known would go a long way to helping the reader, but I'm not sure that would over come the limitations of study design that I perceive here. In general I think the introduction could be fleshed out. There should be some discussion of the different ways of assessing P forms (fractionation, NMR for organic P), and what their pros and cons are. If the ultimate goal is to understand pedogenic effects on P availability to organisms, there is a lot more literature that could be cited. If the goal is to see how P forms vary across this particular "geosequence" then I think more material is need to convince the reader that this a compelling question."

We strongly agree with this statement of Referee 1 that the introduction can be improved. In the revised version of the manuscript, we intend to describe pros and cons of methods that are used to assess P forms in soils. Fractionation, as well as NMR and spectroscopic techniques will be mentioned. We did envisage an "ultimate goal": To refine the model of Walker & Syers (1976) for soils from siliceous parent material (1) by determining the spatial distribution of P in soils with different stage of podzolization, (2) by distinguishing more P fractions, and (3) by examining the relations of different P fractions to important soil properties. In addition, we (4) performed analyses for different soil depths which also supplement existing studies (e.g. Frossard et al., 2000, Ferro Vázquez et al., 2014). At last, also the assessment of the patchiness of different P fractions is novel and has not been described before. A revised discussion will go more into detail on these matters, citing more studies.

"Along those lines, given that rainfall differs by almost 1000 mm/yr between sites, and there are different parent materials, it's hard to understand how these can be considered any sort of sequence. Rather it seems to me that it's four sites that have different

soils, for a variety of reasons that can not be disentangled."

We admit that the term "sequence" is difficult to keep, even though we have already pointed out that the studied soils share a lot of commonalities (Cambisols, silicate parent material, beach forest, and temperate climate). As discussed in our answer to comment (4), in our original manuscript, we ranked our sites into a general concept of P distribution changes during pedogenesis. We propose to introduce the broader, conceptual model of P (fraction) distribution by "stage of podzolization" in a revised version of the manuscript. The soils CON, MIT and LUE will serve as representative examples for early, intermediate and advanced podzolization, respectively. As discussed in the original version of the manuscript, BBR is a special case of early-stage podzolization with a large capacity to withstand podzolization. A revised manuscript will not use the term "sequence of soils" or "geosequence" any more, but "soils with a different stage of podzolization" instead.

"P1 L12 - what is a geosequence? Perhaps better to explain as you did in the introduction, as a series of sites that differ in P status due to differences in parent material and age."

In the revised version of the manuscript, we intend to rephrase the respective paragraphs according to the previous comments on the difficulty to keep the word "geosequence".

"L22 - I do not think that documenting different pools can be translated into an understanding of the pools from which P is acquired. There can be large pools of P that are not useful to organisms on short timescales."

We agree with the Referee that documenting pools cannot straightforwardly be translated into an understanding of the pools from which P is acquired. Other methods, as e.g. P isotope approaches, would be needed. However, enrichment and depletion zones of total P and/or different P pools as identified for the different soils in our study may give hints about low and high P uptake, respectively.

"L24 - Presumably the pedogenesis you refer to is all fairly early stage, and thus P availability is increasing as primary mineral P dissolves."

Not all studied soils have the same pedogenetic age. The BBR soil has formed from basaltic parent material which has been exposed not earlier than in the early Holocene. CON and particularly MIT have formed from gneiss debris which has been weathered intensively under Tertiary tropical climate. In addition, P availability, as characterized by nutrient P contents of beech foliage, is less at LUE (LUE < CON < MIT < BBR, result from pre-studies). The research project, which our subproject is a part of, intends to study the mechanisms that allow temperate forests on silicate bedrock to sufficiently supply the trees with P. It is true that P availability is increasing as primary mineral P dissolves, but the role of pedogenic minerals and its relationship with organic and inorganic P fractions has not been studied in this detail, let alone put in a broader figure of spatial P pool distribution at different stages of podzolization.

"L25 - I am not sure what is novel about this result. The idea that soil development influences P forms and availability is quite old. What is the novel contribution of this work?"

We agree with the Referee that it is well-known that soil development influences P forms and availability. In the revised version of the paper, we intend to clarify the aim and novelty of our study as addressed in previous answers, particularly referring to the issue of spatial distribution of different P forms with different plant availability.

"P2 L19 - How is soil age determined in this geosequence?"

See previous answers; in the revised version of the manuscript, the choice of sampling sites will be rationalized by the different stage of podzolization and not set in a geosequence.

"L22 - Why is the depth distribution important? Are rooting depths different across the geosequence? If so, how?"

Rooting patterns are affected: We have observations of rooting from the soil sampling campaign which we will include in a revised manuscript. However, our methods focused on the spatial imaging of P fractions in soil. In contrast to earlier studies, we also addressed the horizontal variation of P in soil, which might also affect rooting patterns. We demonstrated that and how P accumulates in the studied soils (both in depth and horizontally). This information therefore has implications on plant and microbial studies on these sites.

"L24 - There is a great deal of literature on the distribution of P in soils, though less about P forms. Steven's work in the early 1970s in New Zealand had a wealth of information about P fractions with depth across the Franz Joseph and other chronosequences. The works from Hawaii (Crews et al, 1995 and subsequent) also has information. I believe Paul Selments has P fractions across the San Francisco Volcanic chronosequence, though I can't remember how much depth information he has. And of course Ben Turner has done a lot of work exploring organic N forms in myriad places and across chronosequences."

We failed to clarify that we addressed only temperate forest soils from siliceous parent material. However, we still reckon that the small-scale 2-D depth distribution in the four studied soils is novel information. Moreover, the wet-chemical fractionation provides additional information on different P binding forms, which were related to other soil properties/variables with potential relevance for soil P such as OC and pedogenic Al and Fe minerals. We thank the Referee for the valuable mention of studies. The revised version of the manuscript will also cite these studies in the discussion.

"L26 - I'm not sure from this intro how your data differ from Prietzel et al, 2016b. I'm sure they are different, but your introduction does not set up that difference very well. From my reading of that paper they also looked at P and what it's bound to across these sites."

Our study actually complements the study of Prietzel et al. (2016b), for we were able

to describe the factors controlling P distribution and binding forms (by correlation and factor analysis) at different scales. The approach in this manuscript focused on smaller scales and on distribution patterns within a soil profile (correlation at different depths), whereas Prietzel et al., 2016b focused on P speciation (methodologically), its relationship with parent material and studying soil horizons.

"P3 L8 - Basalt, gneiss and Pleistocene sand are very different parent materials, so I can understand why they would have wildly different P availability, and might host very different forms of P. However, there is no information given as to why the authors suggest these soils are of different age. In Prietzel et al., 2016b it is said that the 10 sites differ in lithology, but I can find no description of how they vary in soil age."

As already described in earlier responses to the Referee's comments, in the revised version of the manuscript the term "soil age", as referred to as "stage of pedogenesis" will be reformulated.

"L13 - This reads as if sampling at each site took place in a single 10cm square. But in truth it took place in a 70x100cm rectangle at every 10cm intersection. Is that correct? If so this text could be clarified?"

The text is indeed not clear enough. The samples were taken in a 70x100cm rectangle at every 10cm intersection. The revised manuscript will clarify this.

"On a more scientific note, why would a single grid be used to get a spatial variation, rather than many different locations? Some more explanation of why this approach was taken is warranted."

In our study, we focused on small-scale spatial variation within a soil profile (cm to m scale, soil profiles). A larger (stand) scale variation study was performed by another group on the same sites and their paper is in preparation. As already mentioned, in the revised manuscript we intend to introduce the importance of small-scale P distribution more pronouncedly.

"P4 - The motivation for the geospatial aspect of this paper is unclear to me. Until coming to the statistical analysis section of the methods, I had no idea there even was a geostatistical analysis, and even at this point in the paper I'm not sure what the goal of such a small spatial scale analysis is. I think this points to the fact that the introduction is so short that it does not really set up the motivation for the study or the questions as well as it needs to to bring the reader along."

In the revised version of the manuscript, we will rephrase the introduction profoundly. Among other issues, we will explain in detail the importance of small-scale variation of P (s. previous answers). The geostatistics, however, provides information about the P and other element's distribution patterns which are difficult to grasp from a numerical data frame. The revised manuscript will furthermore introduce the geostatistical analysis and reason its value to our study earlier.

"P5 L4 - I don't think anyone would expect uniform distributions of P or any of its forms, and in general would expect higher concentrations of total P and organic P in the upper soils. So I'm not sure why this is a major result."

We suggest introducing our results by: "As expected, the interpolated maps..." We tried to report all results from total P down to smaller P fractions and sub-fractions (organic, inorganic, dithionite extractable, oxalate extractable...). The discussion focused on all results, including the higher contents of total P and organic P at the soil profile scale (s. P8 L4-13). We reckon that we not just reported the spatial patterns of higher total P contents, but discussed binding forms and distribution of P in soils with different stages of podzolization, which are novel scientific contributions.

---

## Author Comment (AC2) · 16 Jun 2016

Answers to Reviewer 2:

"1) They estimated "stage of pedogenesis" of their four soil samples. I wonder if it's reasonable to compare among soils with different parent material, moisture content or elevation. How were these four sites selected? Explanation of study sites and detailed soil property data are required. What does "mean" represent (i.e. entire soil depth)? What is the sample number (n=?)?"

In our manuscript, we used the term "stage of pedogenesis" to distinguish the "pedo-genetic age" of our sites (referring to our sites as a "geosequence"). Not all studied

soils have the same pedogenetic age. In the revised version of the paper we intend to replace this ambiguous term using "stage of podzolization" instead to describe the pedogenetic age of our soils. The stage of podzolization is, of course, affected by parent material, climate, elevation and other factors. Our study focused on P distribution in four soils (Cambisols) with different stage of podzolization of forest soils (mainly European beech stands of similar age) with siliceous parent material under temperate climate (as part of the Priority Program 1685 funded by the German Research Foundation DFG). Pre-studies found different P availability, as described by foliar P contents, at these sites (LUE < CON < MIT < BBR). They also found differing P contents in these soils. In the revised manuscript, the soils CON, MIT and LUE will serve as representative examples for early, intermediate and later stages of podzolization in temperate soils with siliceous parent material. As already discussed in the manuscript, BBR is a special case of an early stage podzolization with large capacity to withstand podzolization. A revised manuscript will not use the term "sequence of soils" or "geosequence", but "soils with a different stage of podzolization".

"Mean" represented the mean content in the entire soil depth. A revised version of the manuscript will give the mean contents of the fractions from 0-60 cm depth, to better compare the properties (n = 49 to 56). In addition, we intend to create a new table with detailed information about soil and site properties, then also including stand age, tree composition and soil texture.

"P2 L9: I cannot find any data for LUE in the paper by Prietzel et al (2016b). In addition, the soil property data by Prietzel et al. (2016b) for BBR, CON and MIT were different from their data shown in Table 1. For instance, Prietzel et al. reported pH and TP of MIT (surface 0-2 cm) as 3.8 and 1.99 g P kg-1, respectively, whereas their data were 2.9 and 0.72 g P kg-1, respectively. Some soil properties such as texture and clay content should be added in Table 1."

Prietzel et al. (2016b) reported pH and other soil properties for three of the studied soils by soil horizons. The three soil pH values mentioned by the Referee are all derived from

Ah horizons which have a depth of 10 cm. Likewise, total P values were presented. The pH values shown in this manuscript are derived from 0-5 cm which can affect the estimation. In addition, all soil profiles were freshly prepared for our study which can also complicate comparison of parameters with previous studies from the same site.

"2) They discussed P adsorption mechanisms in acidic soils, yet completely ignored clay content or/and types of clay present in each soil."

This is an important advice and we agree that clay minerals should have been mentioned and discussed in more detail. The fractionation techniques that were used in our study also included dissolution of P adsorbed by clay-sized particles (clay minerals, pedogenic oxides and oxyhydroxides). Our manuscript focused much on Al- and Fe-oxyhydroxides, because P adsorption by clay minerals plays a minor role in P retention. Especially important for P retention by clay minerals is the Al-coverage of clay particles (which generally bind cations in acidic conditions). For most soils, the fraction of oxalate-extractable Al mostly consists of Al cations and Al(OH)3 adsorbed to clay minerals. The latter thus are represented in our assessment of interrelation of P forms with other soil parameters. Moreover, Violante and Pigna (2002) reported that clay minerals (montmorillonite, kaolinite, nontronite, illite, smectite) sorbed less phosphate than poorly-crystalline metal oxides, allophane, mixed Fe-Al gels, organo-mineral complexes, goethite, and gibbsite, because of the smaller surface areas of the latter compounds. In addition, clay minerals mainly exert their influence on P retention by Al and Fe ions which are bound by the silicates. Therefore, the amount and type of Al and Fe oxyhydroxides present in soils is of greater importance than the type of phyllosilicates. Hints on P adsorbed to clay particles are included in the Pdi and Pox fractions, but the fractionation method used cannot distinguish between adsorbed P onto clay minerals and Fe/Al-oxyhydroxides. However, the revised manuscript will include a discussion of the influence of clay minerals and their saturation with different exchangeable cations to P adsorption.

"3) P5 L4:"The interpolated maps did not reveal a uniform distribution of P in any of the

studied soils". This sentence is odd since no one expects uniform distribution of P in soils."

We suggest introducing our results by: "As expected, the interpolated maps..."

"4) I found Figure 5 very confusing. The x-axis indicates, "stage of pedogenesis and soil acidification". However, according to Prietzel et al (2016b), the pHs of the soil samples are in the order of BBR (pH 3.1) > CON (3.6) > MIT (3.8) or according to their data, MIT (2.9) > LUE (3.0) > BBR = CON (3.2). Either way, they are not representing the stage of acidification. As I mentioned, I am not sure if they can compare the stage of pedogenesis among their soil samples."

In the revised version of the manuscript, the term "stage of pedogenesis and soil acidification" will be reformulated into "stage of podzolization". We already argued that the studied soils share commonalities. In our manuscript we ordered our sites into a broader figure of P distribution during pedogenesis. We recognize that it might be difficult to draw the bigger picture of the changes in P binding form and distribution (manuscript Fig. 5) with one replicate per site. However, Fig. 5 and the comparison of the sites should be seen as a conceptual model of relationships between stage of podzolization and the distribution of different P forms and important P fluxes as derived from our results. The discrepancy between the pH values from the publication of Prietzel et al. 2016b and this study has been discussed previously.

"5) Prietzel et al (2016b) estimated $\sim$65% of total P in the upper layer (0-10cm) of BBR was inorganic P, such as $Ca(H_2PO_4)_2$ (11%), apatite (11%) and $FePO_4$ (41%). Also $\sim$40% of total P in the upper layer of MIT was inorganic P, such as $AlPO_4$ (18%) and $FePO_4$ (22%). Yet, Figure 5 shows no inorganic P in the upper layer of BBR or MIT. Any reason why?"

Fig. 5 showed white bubbles with a "P" inside which represented inorganic phosphorus. A revised figure will replace those bubbles further up and also address P adsorption by clay minerals (s. answer to comment 2).

"6) P12 L8: How about adsorption of inorganic P onto clays in the upper layer? According to Prietzel et al. (2016b), the texture of BBR (0-10cm) is silty clay."

We agree that the influence of clay minerals must be discussed (s. answer to comment 2).

"7) P12 L24: Effects of root interaction on P transformation in soils should be included when thinking of distribution of forms of P. It will help to add approx. age of trees in each study site. I imagine that when they collected soil samples, they should have observed plant roots in different layers."

It is true that rooting interacts with the P distribution in soils. We have observations of rooting from the soil sampling campaign which we will include in a revised manuscript. Furthermore, the approximate age of the stands will be included in Table 1.

"8) Figure 2 and 3: I liked the way they showed the distribution patters of TP and different P fractions. However, the range of proportion of each color is not clear. (i.e. what does the range high concentration of P represent?)"

We decided not to insert scale bars for two reasons: 1) the profile images would be far smaller, and 2) we focused on the distribution of P. In the revised manuscript, we intend adding the highest and lowest value of every fraction in the respective image to span the scale. See also example figure (total P at CON).

"9) Table 3: I would like to see actual mean data in addition to the correlation."

We propose compiling a new table (as new Table 2), in which we describe the mean data of the fractions in the full soil profile (0-60cm depth) and in the different compartments (upper, middle and lower profile section). Also n-values will be included.

"10) In conclusion, I suggest adding some sentences to explain how their study can be useful to others and what might be the next step."

We started to address how others can profit from our study by discussing the implications of our results on nutritional strategies (e.g. importance of P recycling). The revised manuscript will work out the benefit for others more clearly (e.g. grid sampling of soil profiles, assessment/evaluation of P bioavailability, implications on bioelement-cycling and ecosystem nutrition strategies, s. Lang et al. (2016), JPNSS, vol. 179 (2), pp. 129-135) and mention next steps, such as studying calcareous soils or including advanced techniques of P speciation (e.g. NMR, XANES).
* * *
[Figure]

[Figure]

**Fig. 1.** Total P at CON, numbers spanning the range of the color pattern